# SMOS brightness temperature assimilation into the Community Land Model

Dominik Rains[1], Xujun Han[2], Hans Lievens[1,3], Carsten Montzka[2], and Niko E.C. Verhoest[1]

[1]Ghent University, Laboratory of Hydrology and Water Management, Ghent, Belgium
[2]Forschungszentrum Jülich GmbH, Institute of Bio- and Geosciences: Agrosphere (IBG-3), Jülich, Germany
[3]Global Modeling and Assimilation Office, NASA Goddard Space Flight Center, Greenbelt, MD, USA

*Correspondence to:* Dominik Rains (Dominik.Rains@ugent.be)

**Abstract.** SMOS (Soil Moisture and Ocean Salinity mission) brightness temperatures at a single incident angle are assimilated into the Community Land Model (CLM) across Australia. Therefore the data assimilation system DasPy is coupled to the Local Ensemble Transform Kalman Filter (LETKF) as well as to the Community Microwave Emission Model (CMEM). Brightness temperature climatologies are precomputed to enable the assimilation of brightness temperature anomalies, making use of 6 years of SMOS data (2010 - 2015). Mean correlation R increases moderately from 0.61 to 0.68 (11%) for upper soil layers if the root-zone is included in the updates. A reduced improvement of 5% is achieved if the assimilation is restricted to the upper soil layers. Root-zone simulations improve by 7% when updating both the top layers and root-zone and by 4% when only updating the top layers. Mean increments and increment standard deviation are compared for the experiments. The long-term assimilation impact is analysed by looking at a set of quantiles computed at each grid cell. Within hydrological monitoring systems, extreme dry or wet conditions are often defined via their relative occurrence, adding great importance to assimilation-induced quantile changes. Although still being limited now, longer L-band radiometer time series will become available and make model output improved by assimilating such data more usable for extreme event statistics.

## 1 Introduction

The potential to improve land surface simulations of soil moisture by assimilating information derived from satellite measurements is well known (Parada and Liang, 2004; De Lannoy et al., 2007; Jia et al., 2009; Chen et al., 2014; Mohanty et al., 2017). Soil moisture products based on data from a number of missions have been used, e.g. ASCAT (Brocca et al., 2010, 2012; Dharssi et al., 2011; Draper et al., 2011), AMSR-E (Reichle et al., 2007; Yang et al., 2007; Draper et al., 2009a) or a combination of both (Draper et al., 2012; Renzullo et al., 2014). Launched in November 2009, the Soil Moisture and Ocean Salinity (SMOS) spacecraft is the first mission specifically designed to map soil moisture from space (Kerr et al., 2001; Mecklenburg et al., 2016), the second one being the similar SMAP mission launched in 2015 (Entekhabi et al., 2010). The passive Imaging Radiometer with Aperture Synthesis (MIRAS) instrument on-board SMOS, sensitive to 1.4 GHz electromagnetic emissions, measures multi-angular top of atmosphere brightness temperatures at horizontal (H) and vertical (V) polarisation. These brightness temperatures are ingested into a complex retrieval algorithm resulting in soil moisture estimates (Kerr et al., 2012) readily usable for analysis, input for higher level products or data assimilation. When assimilating these products, which

roughly represent the top 5 centimetres of the soil column, into the according model layers (Reichle, 2008; Montzka et al., 2012), the assimilation impact in deeper layers will depend on model physics (Montaldo et al., 2001; Kumar et al., 2009; Montzka et al., 2011). Alternatively, by making use of one of the key advantages of the various implementations of the Kalman Filter (Kalman et al., 1960), deeper and unobserved layers can be updated directly. For plants, these deeper layers act as the root zone, where soil moisture has a profound effect on biochemical processes, thus limiting the effect of data assimilation not only to soil moisture (Vereecken et al., 2016). Examples for assimilating SMOS soil moisture retrievals are, among others, given by Martens et al. (2016a), showing that the GLEAM evapotranspiration model can benefit from assimilating these data over Australia, or Lievens et al. (2015b), who conclude that the positive assimilation impact on soil moisture can improve streamflow simulations for the VIC model, as shown in the Murray-Darling basin. The impact on both streamflow and evapo-ration is evaluated by Ridler et al. (2014) for western Denmark. Leroux et al. (2016) assimilate SMOS soil moisture products into the DHSVM model, improving water table depth and streamflow simulations, thereby greatly reducing the uncertainties introduced by the use of uncorrected near real-time precipitation forcings. Scholze et al. (2016) have assimilated SMOS re-trievals together with $CO_2$ measurements to constrain the global carbon cycle.

Apart from assimilating the retrieved soil moisture products, it is also possible to directly assimilate the brightness tem-peratures, which should, in theory, eliminate a number of problems. For instance, the SMOS Level 2 soil moisture retrievals represent the optimum fits between simulated brightness temperatures and the observed satellite signal (Kerr et al., 2012). The simulated top of atmosphere signal thereby depends on both static and dynamic ancillary data based on input and output of a specific land surface model, e.g. for SMOS retrievals the European Centre for Medium-Range Weather Forecasts HTESSEL land surface model (Balsamo et al., 2009). When using a modified or different land surface model, it can be beneficiary to directly assimilate the brightness temperatures in order to use consistent auxiliary information for the land surface model and the radiative transfer model. In the case of assimilating soil moisture retrievals, the auxiliary data used by the model are likely to be correlated with the data used by the retrievals. This inevitably leads to cross-correlated errors between the model and the retrievals, which may have a negative impact on the assimilation performance (De Lannoy and Reichle, 2016a). Some examples of brightness temperature assimilation studies are given by Jia et al. (2009), Muñoz-Sabater (2015), De Lannoy and Reichle (2016a) and Lievens et al. (2016). Taken as a whole, assimilating L-band brightness temperatures in practical terms is quite a new concept which still needs further exploring. The assimilation impact is mostly evaluated by comparing soil mois-ture time series to a limited number of in-situ measurements. Given the proven positive impact and the increased availability of longer time series of satellite observations, hydrological monitoring systems, such as for droughts or floods, are likely to benefit from these data. However, little is still known about long-term assimilation impacts, e.g. on quantiles, which are often used for applications such as drought monitoring.

Within this study, we assimilate SMOS brightness temperatures at H polarisation over Australia from January 2010 until December 2015 into the Community Land Model (version 4.5, Oleson et al. (2013)) and evaluate the assimilation impact both in terms of correlation improvements towards in-situ measurements and in terms of long-term induced model biases, i.e.

changes in quantiles, for the state variable soil moisture. We place the findings within the context of hydrological monitoring systems, which mostly use CDFs as a basis to classify areas of interest. A good overview on the evolution of such hydrological monitoring systems is given by Van Dijk and Renzullo (2011).

We have selected Australia as a study site as we consider it as an ideal test domain for the long-term brightness temperature assimilation. It is quite heterogeneous in terms of climate and largely uninfluenced by human activity, therefore mostly unaffected by Radio Frequency Interference (Leroux et al., 2013). Although large parts are covered by drylands, the land cover varies along the coastline and includes some densely forested areas in the Australian Alps as well as pasture and areas of intense agricultural activity in the south-east and south-west. The lack of large densely vegetated areas, which mask out the

L-band emissions sensitive to soil moisture, is beneficial. Furthermore, soil moisture information based on satellite data is often advertised as being especially useful for monitoring hydrological extremes such as floods and droughts, which Australia is both susceptible to (van Dijk et al., 2013; Johnson et al., 2016; Kiem et al., 2016). In addition to the ones already mentioned, a number of L-band specific studies have focused on Australia, covering soil moisture retrieval (Van der Schalie et al., 2015), assimilation studies (Lievens et al., 2015c), validation studies and field campaigns for SMOS (Peischl et al., 2009; Panciera

et al., 2008) as well as SMAP (Panciera et al., 2014) and soil moisture downscaling experiments (Piles et al., 2011; Merlin et al., 2012; Dumedah et al., 2015). The potential of AMSR-E soil moisture retrievals has been shown by Draper et al. (2009b). A comparison of SMOS satellite soil moisture retrievals with products based on other sensors is given by Su et al. (2013). The joint assimilation of ASCAT and AMSR-E data has been tested by Renzullo et al. (2014). More recently, SMOS soil moisture and GRACE water storage have been jointly assimilated by Tian et al. (2017). Downscaled AMSR-E soil moisture observations

were assimilated within the Murrumbidgee basin by López López et al. (2016).

The Community Land Model (CLM) provides all outputs required for the brightness temperature forward simulations, which further motivates the direct assimilation of brightness temperatures. Being part of the fully coupled Community Earth System Model (CESM), it can be used for future coupled land-atmosphere studies using a similar setup as for the brightness

temperature assimilation. A full description of the CLM surface data used for modelling the Australian continent will be given in section 2.

In order to obtain the brightness temperature forward simulations, the CLM is coupled to the Community Microwave Emission Model (version 5.1, Drusch et al. (2009)) forward operator within the data assimilation system DasPy (Han et al., 2015a). The increments are computed with the Local Ensemble Transform Kalman Filter (Miyoshi and Yamane, 2007; Hunt et al.,

2007). The observation bias between forward simulations and observed brightness temperatures is encountered by assimilating anomalies. Remaining differences in mean and variance are resolved by quantile-mapping the entire observation anomaly time series towards the open-loop forward simulation anomalies at each grid point. Details on the implementation of the assimilation system, the forward simulations and the observation treatment will be given in section 3.

The in-situ data used for the validation are from the OzNet and CosmOZ measurement networks (Smith et al., 2012; Hawdon et al., 2014) and were obtained through the International Soil Moisture Network ISMN (Dorigo et al., 2011). For the quantile analysis quantiles at 1 % steps are computed at each model grid-point, allowing a sufficiently precise empirical estimation of the cumulative distribution functions. To exemplify the effects of quantile changes we show a dry event defined at the 10 % quantile level and to what extent its spatial extent changes when comparing the open-loop run to the data assimilation results. Part of the experiments is also to show how the CLM translates assimilation updates restricted to the upper soil layers into the root-zone purely through model physics as compared to directly updating both the upper soil layers as well as the root-zone, with the findings being set into relationship to the quantile analysis. The results of the experiments will be given in section 4, followed up by the discussion and conclusion in section 5.

## 2   The Community Land Model

The Community Land Model is the land surface component of the Community Earth System Model and can be run offline with pre-computed atmospheric forcings (Oleson et al., 2013). CLM provides global surface datasets which can be interpolated to pre-defined or custom resolutions and grid types both globally as well as regionally, including single point simulations. Interpolating the included surface datasets resulted in artefacts for elevation and grid cell elevation variance as well as plant functional types, with one plant functional type clearly linked to latitudinal borders. We replaced these, but also other surface datasets, with suitable alternatives. For the choice of datasets we kept possible future global applications in mind, which the results of this study could be compared with. At the same time we believe that the Australian continent is well represented by the chosen datasets or that no better suited alternatives were available for the requirements of this study. A description of these datasets is follows in the next section. The model resolution was defined at 0.25 degrees, which agrees well with the Level 3 observations provided in the EASE 25 km grid. The model is run at 30-minute time steps, with hourly outputs, allowing for a sufficiently correct temporal alignment of model and satellite observations.

### 2.1   Surface Datasets

Each grid cell within CLM is divided into land units covering a certain percentage of the total grid cell area. Possible land units consist of vegetation, wetlands, lakes, glaciers and urban areas. Vegetated land units have a single set of soil properties but can be populated by several plant functional types (PFTs), again defined by their percentage of coverage in respect to the entire grid cell (Bonan et al., 2002). We have updated the model PFTs with information from the Moderate Resolution Imaging Spectroradiometer (MODIS) MCD12Q1 (version 5) land cover products, provided at 500 m resolution in sinusoidal projection and containing a classification of each grid cell describing the dominant plant functional type. On the basis of WorldClim climate data (Hijmans et al., 2005) these plant functional types are reclassified to the CLM-compatible PFTs (Bonan et al., 2002). PFTs were then aggregated to the model resolution, computing the percentage of 500 m pixels of each plant functional type per grid cell. Monthly Leaf Area Index (LAI) values for each PFT within a grid cell were computed by averaging the MODIS 8-daily MCD15A3H (version 6) LAI product, also provided at 500 m resolution in sinusoidal projection, over the

assimilation period (2010 - 2015) to derive the monthly climatology and to replace the original climatological LAI values of CLM. The high-resolution LAI values were up-scaled to model resolution by mapping the 500 m pixels to the 500 m reclassified PFT values within each grid cell and subsequently averaging these per PFT. Stem Area Index (SAI) values were also computed on the basis of the high-resolution MODIS LAI data and likewise up-scaled to model resolution, replacing

the standard CLM values. Urban and lake areas were extracted from the MODIS land cover information MCD12Q1. Mean topographic height and standard deviation for each grid cell were downscaled from the 3 arc-second HydroSHEDS digital elevation model (Lehner et al., 2008). Soil texture, namely clay and sand fractions as well as organic matter content, were obtained from the global International Soil Reference and Information System (ISRIC) soil database (Hengl et al., 2014) and mapped to the 10 CLM soil layers by nearest-neighbour interpolation according to their respective depths. The ISRIC database

provided information on organic matter as the gravimetric percentage of the fine scale soil fraction and we assumed that the coarse scale soil fraction contains no organic matter. Bulk density was used to compute the organic matter content required by CLM, assuming 0.58 g organic matter per kilogram. The rational for creating high-resolution datasets for CLM closely followed the approaches described in detail in Ke et al. (2012) and Han et al. (2012), who similarly replaced the CLM standard datasets.

**2.2 ERA-Interim atmospheric forcing**

CLM provides forcings (CRUNCEP) which do not cover the required time period. Therefore, due to the release of ERA-Interim reanalysis data (Dee et al., 2011) with a time-lag of only a few months, these data were used to force the CLM land surface model over Australia.

The variables 2 m air temperature, 2 m pressure, short-wave incoming radiation and total precipitation were extracted and

specific humidity was computed from the ERA-Interim 2 m dew point temperature and 2 m air temperature. 2 m wind speed was derived from the provided wind speed components in lateral and longitudinal direction. With ERA-Interim being produced by assimilating a multitude of observations into an atmospheric model, some of these variables are the result of the analysis step and others of the forecast step, thus the data needed to be handled respectively. Forecasts for flux variables are provided bi-daily at 0:00 and 12:00 UTC for 3, 6, 9 and 12-hour forecast periods and in accumulated form. For example, the precipitation

forecast for a 6-hour time window is the accumulated precipitation over 6 hours. In order to obtain a precipitation estimate for the hours 3 - 6, the precipitation forecast for the first 3-hour window needs to be subtracted. This disaggregation was performed for all flux variables to obtain 3-hourly forcing estimates. Analysis variables are valid as instantaneous estimates and no disaggregation had to be performed in their case. The atmospheric forcings were bi-linearly interpolated from 0.75 degrees spatial resolution to 0.25 degrees model resolution. A similar approach for creating atmospheric forcing data based on

ERA-Interim, but with additional corrections through ancillary data, is described in Weedon et al. (2014). Time interpolation from 3-hourly to 1-hourly timesteps is performed at CLM runtime with an appropriate interpolation algorithm applied to each variable. Incoming radiation is interpolated by using a cosine function simulating the position of the sun, for precipitation a nearest neighbour interpolation is used. For the remaining variables linear interpolation is applied.

## 3 Assimilation system

The assimilation experiments are performed with the open-source multivariate data assimilation system DasPy. Mainly coded in Python, its modular design in principle allows the coupling of different models, observation operators and assimilation algorithms. The version used within this study is coupled to the Community Land Model and the Community Microwave Emissions Model (CMEM, de Rosnay et al. (2009)) observation operator. Furthermore, the system uses the Local Ensemble Transform Kalman Filter (LETKF) implementation by Miyoshi and Yamane (2007) for computing the actual increments. Several studies have been performed using DasPy, including the assimilation of synthetic brightness temperatures within the Babaohe River Basin in northwestern China (Han et al., 2012) and in the Rur catchment in Germany (Han et al., 2015b). The system allows for dual state parameter estimation as shown in Han et al. (2014b).

DasPy has been developed with a focus on High-Performance Computing. Parallelism is achieved through ParallelPython, OpenMP, the Message Parsing Interface (MPI) and MPI4Python. Ensemble members can be distributed across different nodes with the core assimilation system, including the LETKF, being confined to one node. Some of the operations are implemented in C++ within the Python environment, using Weave, to further optimise performance. The LETKF itself is a fully parallel Fortran implementation called through F2PY (Fortran2Python).

### 3.1 Local Ensemble Transform Kalman Filter

The Local Ensemble Transform Kalman Filter (Hunt et al., 2007) is one of the implementations of the Ensemble Kalman square root filter and is deterministic as opposed to stochastic, thus not introducing random noise into the observations. The LETKF has the advantage over other non-localised implementations that the analysis performed for each grid point is limited to a local domain, which makes it computationally more efficient and less susceptible to long-range spurious correlations. The original SMOS footprint is 43 km across and thus covers more than a single model grid cell, which would justify the assimilation in 3D. However, mostly for reasons of simplicity, and also due to the previously performed inverse distance observation regridding partially accounting for this, we only use observations directly covering a grid cell. Also, about 90 % of the signal observed by SMOS does originate from a footprint closely matching the model grid.

Mathematically, the LETKF can be described as follows: Model states for each ensemble member $k$ from a total of $K$ ensemble members are propagated over time by the model $M$, starting at a previous analysis time step $n-1$, e.g. a previous analysis step within the data assimilation scheme, $x^a_{n-1}$. This results in a new background estimate of the state vector $x^b$ consisting of the soil moisture states for all ensembles at the current time step $n$.

$$x^b_{n,k} = M_n(x^a_{n-1,k}) \tag{1}$$

The background ensemble perturbations $X^b$ at the current time step can be computed as:

$$X^b = [x^b_1 - \bar{x}^b|...|x^b_k - \bar{x}^b] \tag{2}$$

The individual ensemble states $x^b$ are mapped into observation space using a forward operator $H$, in this case CMEM.

$$y^b_k = H(x^b_k) \tag{3}$$

and the forward simulation perturbations are defined as:

$$Y^b = [y_1^b - \bar{y}^b | ... | y_k^b - \bar{y}^b] \tag{4}$$

Within the ensemble space the analysis error covariance $\tilde{P}^a$ is computed through

$$\tilde{P}^a = [(K-1)I + (Y^b)^T R^{-1} Y^b]^{-1} \tag{5}$$

allowing for the computation of $\bar{W}^a$ as the mean weighting vector

$$\bar{w}^a = \tilde{P}_a Y^{bT} R^{-1} (y^0 - \bar{y}^b) \tag{6}$$

resulting in the analysis mean $\bar{x}_a$.

$$\bar{x}_a = \bar{x}_b + X^b \bar{w}^a \tag{7}$$

The analysis perturbations are defined as $X^a$

$$X^a = X^b + W^a \tag{8}$$

with

$$\tilde{W}^a = \sqrt{(K-1)\tilde{P}^a} \tag{9}$$

where the analysis error covariance $P^a$ is given by:

$$P^a = X^b \tilde{P}^a (X^b)^T \tag{10}$$

## 3.2 Ensemble generation

Model uncertainty is simulated by running the model in ensembles with perturbations applied either to the atmospheric forcings, surface dataset, model parameters or possible combinations of these. In order to account for the model uncertainty in this study, CLM is run with 32 ensembles with spatially-uncorrelated perturbations added to some of the ERA-Interim forcing data, namely air temperature, shortwave radiation and precipitation. Shortwave radiation is perturbed with multiplicative noise with a standard deviation of 0.3, whereas for temperature additive noise with a standard deviation of 2.5 K is applied. Finally precipitation is perturbed with multiplicative log-normal noise with a standard deviation of 0.3. The perturbation factors are the same as used by Reichle et al. (2007) and Han et al. (2014a). No spatially correlated noise was added, as the experiments are carried out in 1D, only using one observation per grid cell. To avoid ensemble collapse during dry periods, soil texture is also perturbed once at model startup. Here, multiplicative noise with a standard deviation of 10 percent for clay and sand for the top two soil layers is applied. For lower layers the top layer multiplicative factor is rescaled by using the inverse relationship between the thickness of each soil layer and the summed soil layer thickness of the two top layers (see Table 1). This is to ensure that increments in lower soil layers do not result in very large changes in soil water in absolute terms, since soil layer thickness greatly increases towards lower layers. With CLM deriving hydraulic properties based on soil texture, it is to be noted that as a consequence each ensemble member runs with slightly modified model physics. Concerning the number of ensembles, an amount of around 30 is common in brightness temperature assimilation studies and should allow sufficient error estimation.

### 3.3 Observation Operator

Forward simulations from the model space to the observation space are performed with the Community Microwave Emission Model (CMEM, version 5.1). Model output at each observation time, with the observation time rounded to the full hour, serve as input in order to simulate brightness temperatures as measured by the satellite. SMOS ascending and descending orbits have a local overpass time of approximately 6 am and 6 pm. Forward simulations are thus computed at 08:00 UTC on the same day and 20:00 UTC on the previous day for descending and ascending acquisitions respectively, assuming an average time shift of -10 hours for the entire Australian continent. This greatly decreases the number of analysis steps, since individual orbits within one day can be assimilated at once, assuming that a sufficiently correct temporal alignment between observations and model forward simulations is provided. With the western parts of Australia deviating by 2 hours and the ERA-Interim forcings being interpolated from 3-hourly to 1-hourly data, we consider this approach to be acceptable.

CMEM requires time invariant information such as soil layer depth, sand, clay and water fractions, surface height as well as the dominant vegetation type covering the grid cell. Plant functional types are reclassified to ECOCLIMAP vegetation classes (Champeaux et al., 2005) and the type for low and high vegetation is then used by the CMEM. Based on this reclassification, the LAI information is assigned to the ECOCLIMAP low vegetation classes accordingly. For the offline forward simulations CLM was run with LAI as daily output in order to make use of the model-internal LAI interpolation, creating a smooth LAI time series based on the monthly surface dataset. This also ensures that the LAI values used for the CMEM forward simulation are the same as those used within CLM during assimilation. LAI values for high vegetation classes are fixed within CMEM and not taken from the CLM input data. Other dynamic fields used in the forward simulations are soil moisture and soil temperature for all defined soil layers and 2 m air temperature. CMEM supports different types of sub-modules for specific calculations. Within this study, the Mironov model (Mironov et al., 2004) has been chosen for the dielectric constant computation. Vegetation temperature is computed directly by CLM and used as an input without the need of an approximation, e.g. through air temperature. Effective temperature is obtained through the Wigneron model (Wigneron et al., 2001) and applied in the dielectric model. For smooth surface emissivity, soil roughness and vegetation opacity, the Fresnel, Choudhury (Choudhury et al., 1979) and Wigneron (Wigneron et al., 2007) models are used respectively. Finally, atmospheric contributions are estimated with the Pellarin methodology (Pellarin et al., 2003). For all modules the standard parameters for CMEM 5.1 remained unchanged and the forward observation model was not calibrated. Although the standard parameters are very unlikely to be perfect for the different land cover classes, we argue that this approach is not necessarily worse than the alternative of calibrating the radiative transfer model. By modifying parameters, such as surface roughness, the bias between forward simulations and observations can be removed, but in some cases at the expense of a reduced sensitivity towards soil moisture. Therefore we remove the static bias between simulations and observations through observation rescaling.

### 3.4 Observations and anomaly preparation

Large biases are common between modelled and observed brightness temperatures due to the many uncertainties involved, such as in the atmospheric forcing, the land surface representation, the land surface model itself as well as the radiative

transfer model and its parametrization (Drusch et al., 2009; Barella-Ortiz et al., 2015), with this study being no exception. The assimilation is expected to correct random errors only, i.e. bias-blind, and it is therefore necessary to remove the bias prior to assimilation (Yilmaz and Crow, 2013). Calibrating the radiative transfer model to closely match the observed time series is a possible solution, as shown by Drusch et al. (2009), De Lannoy et al. (2013) and Lievens et al. (2015a), with the alternative being the rescaling of the measurements to mimic more closely the forward simulations (Lievens et al., 2015b), as mentioned above. The details of preparing the observations prior to assimilation are given here.

SMOS Level 3 daily brightness temperatures at horizontal H polarisation and 42.5 incidence angle provided by Centre Aval de Traitement des Données (CATDS) are used in the study and processed for the years 2010 - 2015 (version 310). The data are rigorously filtered by using ancillary data from the corresponding Level 3 soil moisture products (version 300), excluding measurements with a probability of Radio Frequency Interference (RFI) greater than 0.2 and a Data Quality Index (DQX) value greater than 0.07. Measurements with a number of activated science flags, namely strong topography, snow, flooding, urban areas, coastal zone and precipitation, are not considered either. The filtered observation data are regridded from the Equal-Area Scalable Earth Grid 2 (EASE2) 25 km grid to the 0.25 degree rectangular model grid by using inverse-distance interpolation.

On the basis of these data, we compute the climatology for each day for the years 2010 - 2015 by averaging along a 7-day moving window across the 6 years, producing separate climatologies for ascending and descending orbits, thus removing seasonal differences between forward simulations and observations (see e.g. De Lannoy and Reichle (2016a)). Anomalies are then computed between the climatologies and the original SMOS time series. Brightness temperature forward simulations based on an open-loop run with 32 ensembles are performed and the ensemble mean climatology is derived in the same way as the observation climatology. SMOS anomalies are then quantile-matched to the ensemble average forward simulation anomalies to account for the differences in variance. The full approach of anomaly computation and quantile matching is to account for seasonal mean differences between simulations and observations and to remove the bias without more aggressive CDF-matching techniques at seasonal level being required. The original brightness temperature simulations over the entire period exhibited a mean warm bias of 21 K for the ascending orbit and 26 K for the descending orbit. Anomaly correlations prior to quantile matching are 0.21 and 0.39 and after quantile matching 0.38 and 0.60 for ascending and descending orbits respectively. Based on the scaling factor between the standard deviation of the original and CDF-matched SMOS anomalies, the observation variance is recomputed. The unscaled observation variance R = 5 $K^2$ was defined, accounting for a standard instrument error of 3 K and an assumed combined standard mean error of 4 K for the forward simulations and representativeness error. The instrument error can be seen as a low estimate and is based on the assumption that the brightness temperature binning around the 42.5 degree incidence angle results in a slight reduction, when compared to the 4 K instrument error usually applied for Level 1 data.

During assimilation at each time step the current forward simulation is subtracted from the precomputed forward simulation climatology to compute on-the-fly anomalies. The difference between this simulated anomaly and the SMOS anomaly is the innovation, which is used within the LETKF algorithm. The assumption is made that the forward simulation climatology does not significantly change during the assimilation run. In total there are 2063 and 2044 observations for the ascending / descending orbits respectively.

## 4  Data assimilation and results

In total, three assimilation experiments are carried out, updating either top layer soil moisture or both top layer and root-zone soil moisture. Only updating the upper soil moisture allows for testing the ability of the model to feed the assimilation effects into the root-zone through model physics only. Updating the root-zone is carried out with two sets of soil texture perturbations,

which largely influence the modelled background error. The objective is to validate the assimilation impact by comparing the time series before and after assimilation to a number of in-situ measurements. In addition, shifts in the soil moisture quantiles in respect to the open-loop run are analysed to highlight some long-term effects of the data assimilation. A set of quantiles is computed at each grid cell to allow the empirical estimation of the cumulative distribution functions, since shifts, both positive or negative, are possible at different quantile levels. Both the impact on correlation and the long-term effects are set into

relationship to which layers are updated and to the model background error in the root-zone. The experiments and their results are described in the following and set into context of their potential effect on hydrological monitoring systems, as shown for the exemplary classification of a dry event. We believe this to be relevant, since L-band data, or data from other sources, are in the long run likely to be incorporated into more and more operational systems.

The spatial patterns of the open-loop soil moisture simulations at different depths were compared to the locally optimised

AWRA-L land surface model (http://www.bom.gov.au/water/landscape) to ensure that the CLM simulations are plausible.

In the first experiment (DA 1) only the upper three CLM soil layers, corresponding to a depth of 9 cm, are updated. Although the brightness temperatures are only sensitive to soil moisture in up to 5 cm depth, DA 1 was defined as updating the top three layers, since a number of in-situ measurements are taken from a depth of up to 8 cm. For these in-situ sites, measurements are also available for deeper layers and we thus define top layer soil moisture as the soil moisture updated in DA 1. The upper six

model layers, reaching 50 cm soil depth, are updated in the second experiment (DA 2). We refer to the lower three of these soil layers as the root-zone. These two experiments enable us to examine to what extent CLM model physics alone are sufficient to update the root-zone through the effects of the assimilation on the upper layers, as in comparison to directly applying the increments in this depth. For the experiments DA 1 and DA 2, soil texture perturbations were incrementally reduced with layer depth, minimising the impact of potentially large updates in deep layers. Since increments are computed in relative

soil moisture, identical increments affect absolute soil water very differently, greatly exaggerating the assimilation impact for deeper layers. Perturbations for the two top layers remain unchanged, thus not decreasing the ensemble spread for the layers where SMOS is sensitive to soil moisture. The soil texture of the subsequent layers is perturbed by decreasing the perturbation factor by the inverse ratio between the respective layer thickness and the layer thickness of the two top layers (see ensemble generation under section 3 and Table 1). Within a third experiment (DA 0), homogeneous soil texture perturbations are applied

across all layers, highlighting the problem of large increments in lower layers. As will be shown, the larger ensemble spread in DA 0 actually further improves the correlation with in-situ measurements, but at the expense of introducing strong long-term effects. For all experiments the brightness temperature forward simulations are computed by using the CLM output of all layers. The L-band simulations are thereby mostly affected by the output of up to 5 cm depth, which corresponds to the sensitivity of the SMOS sensor.

## 4.1 Correlation with in-situ observations

For validation, hourly CLM soil moisture output is compared to in-situ measurements obtained from the International Soil Moisture Network ISMN (Dorigo et al., 2011). OzNet in-situ measurement probes are located within the Murrumbidgee catchment in south-east Australia, a limited spatial domain which does however cover a range of different land cover classes representative for Australia (Smith et al., 2012). The Murrumbidgee catchment was also chosen as a site for a SMOS validation campaign (Peischl et al., 2012). Measurements within the OzNeT network are taken with TDR-probes at shallow levels, mostly 5 cm or 8 cm, and at deeper layers, mostly 30, 60 and 90 cm. The in-situ measurements that are part of the CosmOZ network are taken by using cosmic-ray neutron probes and are therefore representative for a larger horizontal footprint than the more traditional measurements. CosmOz measurement sites are located within the Murrumbidgee catchment as well as at selected locations close to the Australian coast. In addition to the original description of the measurement networks Renzullo et al. (2014) and Holgate et al. (2016) for instance offer an extensive overview of the CosmOz and OzNet measurement sites. Su et al. (2013) give more details on the Murrumbidgee catchment and the locations of the OzNet measurement sites. For all in-situ measurements sites the weighted average of the corresponding CLM soil moisture layers is taken, with the layer thickness being used as the respective weights. Figure 1 shows where the Murrumbidgee catchment is situated as well as the land cover data used for the CLM simulations.

Taking into account only measurements with at least one year of data, not necessarily consecutive, correlations improve from 0.613 for the open-loop run to 0.640, 0.678 and 0.681 for DA 1, DA 2 and DA 0 for top layer soil moisture (number of stations n = 17). Root-zone soil moisture improvements are smaller, with average correlation coefficients of 0.626, 0.644 and 0.648 for DA 1, DA 2 and DA 0 compared to 0.601 for the open-loop run (n = 31). For the upper level soil moisture, correlation improves for all in-situ measurement stations, whereas for the root-zone soil moisture a single in-situ station shows a deterioration of correlation for DA 1. In the case of DA 2 and DA 0 the correlation at two stations, albeit at different ones, slightly deteriorates. On average, upper soil moisture behaviour thus improves by additionally updating deeper layers, whereas deeper layer soil moisture is slightly enhanced through only updating top level soil moisture, with the assimilation effects only being applied through model physics. All in all, updating the top six CLM layers results in the largest improvements, even more so if the identical soil texture perturbations are applied to all soil layers within experiment DA 0, thereby increasing the assimilation impact through an increased ensemble spread and background error. The individual in-situ measurements around the area of the Murrumbidgee catchment and the respective correlation changes for all three experiments towards the open-loop run are shown in Figure 2. For top layer soil moisture the largest improvements are visible for the sites located in the centre of the catchment (Yanco site) with clear improvements for DA 2 and DA 0 when compared to DA 1. In the case of the root-zone, multiple measurements at different depths were averaged using the CLM layer thickness as weights. Here improvements are also highest for the Yanco site, except for one measurement location showing a deterioration of correlation for DA 0. The area around the Yanco site is flat and semi-arid with mostly low vegetation and thus more ideal for L-band soil moisture sensitivity. The lesser improvements for the other in-situ sites towards the east therefore could be explained by the more complex terrain, less homogeneous soil texture and higher vegetation influencing the L-band signal, as discussed by Su et al. (2013).

Figure 3 shows The Taylor diagrams for the in-situ validation of experiment DA 2. As opposed to Figure 2, all original measurements are included with no vertical aggregation performed. The Taylor diagrams reveal a slightly decreased normalised standard deviation when compared to the open-loop time series. In terms of standardised RMSE it is less conclusive, with RMSE being slightly reduced for some stations and slightly increased for others, but never significantly. These findings correspond well to experiments DA 0 and DA 1 (not shown).

When comparing the average changes in correlation for the non-vertically aggregated sites only for the ten used CosmOz sites, correlation increases by about 0.016 for DA1 (three sites with +0.03 and two very close to zero). For DA2 the average correlation increases by 0.02 (two sites showing +0.05 and one -0.02) and for DA3 by 0.13 (three sites slightly deteriorating). This highlights that the assimilation improvements are stronger for the OzNet sites located in the Murrumbidgee catchment. Partly this might be attributed to the fact that the CosmOz measurements are valid for a variable soil depth, depending on the current soil moisture conditions. For validation a single soil depth was used, which is reported to the ISMN network. Also, CosmOz sites are partly situated along the coast or close to water bodies and within areas of higher vegetation, making improvements through data assimilation more challenging, as reported by Renzullo et al. (2014). When only considering the CosmOz sites, correlation decreases for DA3 in respect to DA2 which contradicts the findings when taking all measurements into account.

Altogether, the results demonstrate that the assimilation system has been sufficiently well designed to improve the modelling of soil moisture for the Australian continent, both for top layer soil moisture and the root-zone. However, as with most assimilation studies, validation sites are sparse and do not cover the many-fold combinations of soil texture, land cover, climate etc. which might all have an impact on the assimilation performance. The representativeness error of the in-situ measurement equally remains a problem, with the spatial support of the measurements, in the case of TDR probes only point measurements, being smaller than the area covered by satellite. The assimilation system is therefore not designed to remove the relative bias between soil moisture simulations and observations, since the exact truth remains unknown, but to improve the temporal behaviour of the simulations, which has for the most part been achieved.

## 4.2    Soil moisture increments

Especially over long time periods the mean increments in a bias-blind assimilation system can be expected to be very close to zero. Figure 4 and Figure 6 show the mean soil moisture increments over the assimilation period 2010 - 2015 for the experiments DA 0, DA 1 and DA 2, separately for the ascending and descending orbits. Both the top three soil layers and the root-zone soil layers were averaged. Distinctive areas of mean positive increments for the ascending orbit are visible in the north, south-west and south-east of Australia, seemingly being linked to the occurrence of vegetation (see Figure 1). The areas in the south-west and south-east as well as in the north correspond well to the subtropical, temperature and tropical climate zones respectively and are subject to higher precipitation than the dry areas inland, although other areas along the coast have similar precipitation. The areas in the south-west and south-east correspond to the wheat growing areas of Australia. The misrepresentation of these areas through the CLM surface datasets, such as the use of climatological LAI instead of actual LAI, might well be the source of such patterns. Also, satellite-based estimates of Leaf Area Index are not error-free, as has

been shown specifically for the Murrumbidgee catchment (McColl et al., 2011). Irrigation, which is predominantly applied within the south-east, could be an additional source of error for limited areas, since the forward simulations will be based on seasonally too low soil moisture, causing an incorrect estimation of the brightness temperature seasonality and the subsequent anomaly computation.

Nevertheless, for the ascending overpasses the positive biases hardly exceed 0.5 % soil moisture and the remaining parts of Australia either show no mean increments bias or slightly negative values, both in areas covered by mostly sparse vegetation and the inner drylands. For the descending orbits the patterns are still visible but are weaker, both for top layers and the root-zone. The exception is DA 0, where little differences between top layer and deep layer increments are noticeable. Interesting to highlight is the fact that top layer deviations from zero are strongest for the assimilation experiment DA 1, compared to DA

0 and DA 2, which both update top layer soil as well as root zone. The reason may be that updating deeper layers results in a more lasting effect, moving the model closer to subsequent observations and thus reducing subsequent increments.

Figure 5 and 7 show the increment standard deviations. Spatial patterns of the assimilation impact are very distinctive but do not necessarily correspond to the patterns seen in the mean increments, although they do partly match, as for the south-west and south-east. The relatively large increments in the western Wheatbelt and the Murray-Darling basin, some areas close to

the western coast of Queensland and the eastern coast of the Northern Territory show a standard deviation of 2.5 %. The areas in the north seem to be consistent with the occurrence of tussock grasses, as shown by the Australian National land cover map (http://www.ga.gov.au/scientific-topics/earth-obs/). Minimal or zero increments in all layers, especially along the eastern coast, are due to a lack of observations, as these were removed due to the active vegetation science flags or the fact that the high LAI values for high vegetation prescribed within the forward operator mask all signals from soil moisture.

Concerning the different assimilation experiments, top layer increments are largest for DA 1, followed by DA 2 and then DA 0. Being the most dominant dynamic factor for the ensemble generation, precipitation leads to an immediate increase in ensemble spread and, as a consequence, to a larger background error for the very shallow soil layers, thereby also increasing the observation impact. This impact on the ensemble spread will however be dampened and temporally lagged for deeper layers. With increasing layer depth, the soil texture perturbations play a more important role in determining the background error.

This is visible for DA 0 where homogeneous soil texture perturbations were applied across all layers and increments in the root-zone are not significantly smaller than for the layers above (Figure 5). As a contrast, root-zone increments applied within experiment DA 2 are far smaller than for the upper layers. For the validation with in-situ measurements we showed that these larger increments for DA 0 actually result in a slightly increased correlation over DA 1.

Concluding on the behaviour of increment bias and increment standard deviation, it seems that there is a relationship to root-

zone updates. Both increment bias and increment standard deviation are largest for DA 1, where the root-zone is not updated at all. The top layer increment standard deviation decreases for DA 2, whilst also updating the root-zone, with a slight decrease of the increment bias. Compared to DA 2, the increment standard deviation is larger in the root-zone for DA 0 and the top-layer increment bias decreases substantially.

As for the differences between the ascending and descending orbits, we conclude that they can be partly explained by

referring back to the fact that soil moisture retrievals are expected to be of a higher quality for the ascending orbits (Hornbuckle

and England, 2005; Kerr et al., 2010). The thermal equilibrium within the soils, which is more pronounced at 6 a.m. local time, reduces the error in the forward simulations. The mean bias between the forward simulations and observations is 5 K less for the ascending orbits, which supports this explanation.

To highlight some of the seasonal effects, Figure 8 shows the increment standard deviation exemplary for DA 2 and the ascending orbit both for the months January to March and June to August. For the austral winter, increments are largest for the agricultural areas in the south-west and south-east, now in the growing season, and these seasonal effects clearly dominate the average of the increment standard deviation (compare to Figure 7). Similarly, the patterns in the north, mostly linked to grassland, are visible in the yearly average and the shown months contribute the most to their existence. Differing seasonal effects of the assimilation impact were also observed by Martens et al. (2016b) and Tian et al. (2017), although the observed patterns are distinctively different.

When comparing the winter patterns to areas where irrigation takes place, as shown by van Dijk et al. (2013), irrigated areas within the Murray-Darling basin can be identified through an increased increment standard deviation. Here the SMOS observations correct soil moisture dynamics which are not explicitly modelled. Kumar et al. (2015); Escorihuela and Quintana-Seguí (2016) and De Lannoy and Reichle (2016b) have similarly reported on the potential of SMOS to observe irrigation.

## 4.3   Soil moisture quantiles

Apart from looking at the increments, we compute a set of quantiles at 1 % intervals for each CLM soil layer and each grid point, both for the assimilation experiments and the open-loop run. Although in principle the assimilation system should be designed bias-free with similar positive and negative increments, the previous section has revealed that small increment biases do exist, potentially causing long-term effects in the resulting analysis. Figures 9, 10 and 11 show the 10 % quantile changes, thus very dry conditions, in relation to the open-loop run for the top nine CLM layers. For experiment DA 1 (Figure 10), assimilation has an impact on the topmost soil layer with the quantile increasing by by a maximum of ca. 1 % for large areas and by up to 4 % for spatially very limited areas. Much smaller changes are visible for the second and third layer, with some areas also showing a negative impact by up to 2 %. CLM model physics result in changes being also visible within the root zone, CLM layers three to six, and below. One of the visible patterns is again south-east Australia. For the very deep layers some independent patterns emerge which are not visible for the above layers. Most notably in the Nullarbor plain, on the south coast of Australia, where the 10 % quantile increases by up to 2 %. Such patterns are related to strong singular increments in very dry areas which accumulate in the deepest layers. Due to the low temporal dynamics in these lower layers, any added water will have a lasting effect especially on lower quantiles. For experiment DA 2 (Figure 11), with the root zone also being updated, larger impacts on the quantiles in deeper layers can be observed. For the most part the patterns reflect well the ones identified for DA 1. Figure 9 shows the impact on the 10 % quantile for experiment DA 0 with homogeneous soil texture perturbations being used. As expected, significant effects are visible especially within the root zone, with quantiles being decreased over wide areas of the Australian inland by up to 5 %. This is the result of the mean increments being slightly negative for inland Australia, which has a large effect when allowing large updates. Also, since absolute soil moisture increases with layer depth due to increasing layer thickness, removal of water in low layers increases drainage in the above layers, resulting in these to

dry out. This is especially visible for layers two and three, where inland Australia to a far greater extent shows a lowered 10 % quantile in comparison to DA 1 and DA 2. For the lowest layer a clear positive quantile shift is visible in the area of Lake Eyre. The land cover map in Figure 1 shows this as the only area that is classified as bare soil, although it is mostly a salt plain with water levels of the lake itself being strongly seasonal. A number of observations were therefore flagged, making the computation of a stable brightness temperature climatology challenging.

Figures 10 - 11 focus on the changes of the 10 % quantile. However, the spatial patterns identified do not necessarily reflect changes at other quantile levels. The complex nature of these shifts throughout the entire CDFs is shown in Figure 12. The continental average empirical cumulative distribution functions are plotted for soil layers 1 - 6 for the open-loop run as well as for DA 1 and DA 2. Lower quantiles are increased on average through data assimilation, although at extremely low levels the behaviour tends to reverses again. Here the quantiles decrease when compared to the open-loop. For the upper quantiles a small decrease can also be observed. The point where the decrease turns into an increase, with the assimilation having an on average neutral impact, is roughly the 50 % quantile for the top layer. For the subsequent layers this point decreases towards the 40 % quantile. Although DA 0 resulted in the best correlation with the in-situ measurements, it was disregarded at this point since the assimilation impact was too disruptive by strongly drying out the model across many layers.

For DA 1 and DA 2, the interplay of the quantile changes at the various levels results in an average decrease of the standard deviation of the soil moisture analysis, which to a certain extent could be attributed to the anomaly rescaling. First of all, due to the low sample count any quantile-mapping procedure tends to be a challenge around the extremes of the distributions. Additionally, the exact observation error for each observation is unknown and although expected to be zero on average, with a small sample size the observation error might on average deviate from zero, affecting the rescaling, since the true limits of the observation anomaly CDFs are unknown. However, many other reasons will equally play a role and as it has been shown, increment bias and standard deviation are linked to certain geographic areas. The disentanglement of the desired systematic enhancements from erroneously introduced effects remains a challenge. There is still no perfect approach to rescaling the observations to match the model or to calibrating the forward observation model. Looking at long-term CDF changes induced by assimilation can be part of evaluating these different approaches with the final application of analysis data to be kept in mind.

Finally, as an example we want to place quantile behaviour within the context of possible hydrological monitoring systems which directly make use of grid cell quantiles and empirical CDFs. The correction of short-term behaviour alone, i.e. hourly or daily, has a minor effect when analysing phenomena that spread across larger spatial scales and time intervals, although large increments, e.g. due to corrected precipitation during a storm, can have an effect on the start and end point of an observed phenomenon, such as a drought. When classifying such an event defined at a specific quantile level, there will be a twofold impact from the assimilation: the changes in the quantile of interest as well as the change in soil moisture itself. Here we highlight a sample dry event on the east coast to show to what extent its classification changes through the assimilation impact. Figure 13 shows root-zone soil moisture at or below the 10 % quantile level for the open-loop run as well as the data assimilation experiment DA 2 for soil moisture conditions in early 2010, thus at the beginning of the assimilation period. Due to the higher 10 % quantile for DA 2, as seen in Figure 11 and 12, the spatial extent of the cluster for DA 2 is reduced, but the spatial patterns

of soil moisture remain largely the same. At some time periods, not shown here, a higher degree of noise is noticeable within the assimilation dataset. This is likely due to the fact that non-spatially correlated noise was applied to the meteorological forcings, resulting in a heterogeneous background error field for grid points. We thus conclude that despite having carried out the assimilation in 1D, spatially correlated noise is recommended for such applications. An alternative would be to further increase the ensemble size, but at the expense of higher computational resources. Additionally, when trying to extract meaningful statistics on the occurrence of events, such as droughts, it might be particularly important to clean up the dataset in the case of data assimilation using simple filter algorithms, such as applied by Herrera-Estrada et al. (2017). We want to highlight the fact that the shown event is for demonstration purpose and not linked to any major drought event, which would require a more in-depth analysis and references to independent data sources.

## 5  Discussion and Conclusion

The Community Land Model was set up for the Australian continent and coupled to the Community Microwave Emission Model. We have substituted the surface datasets with higher resolution and more recent data. Additionally, we have replaced the offline forcings with the ERA-Interim reanalysis. The assimilation over 6 full years, from 2010 – 2015, of SMOS brightness temperature anomalies with the LETKF improved soil moisture simulations when compared to in-situ measurements in the order of up to 11 % for top soil moisture. Both the CLM model and the forward observation model were not calibrated, therefore implying that the assimilation system could be applied to other areas.

In detail, three data assimilation experiments were carried out: Within the first experiment the top three layers were updated, which mostly correspond to the depth where SMOS is sensitive to changes in soil moisture and top layer in-situ measurements are available. The correlation with top layer soil moisture measurements increased by 5 %, root-zone soil moisture increased by 4 %. Within the second experiment both top soil moisture and the root zone were updated, resulting in correlation improvements of 11 % and 7 % respectively. The CLM is therefore able to translate top layer updates into deeper layer soils. Greater improvements can be achieved by additionally updating the root-zone directly. For these two experiments, soil texture perturbations were reduced with increasing layer depth. With CLM layer thickness vastly increasing with depth homogeneous soil perturbations across all layers result in large deep layer updates in terms of absolute soil moisture. This was demonstrated in a third experiment, where correlation with in-situ measurements was highest compared to the first two experiments, namely 11 % and 8 % for top and root-zone soil moisture respectively. This coincides with the findings by Kumar et al. (2009), who report that soil moisture simulations profit more from assimilation with an exaggerated coupling between top-layer and root-zone soil moisture than vice versa. Within this context we interpret the overly large root-zone updates for the third experiment as an artificially exaggerated coupling. Kumar et al. (2009) also state that when compared to other land surface models, CLM actually shows an overall lower coupling strength. Larger improvements in root-zone soil moisture simulations therefore might be possible when using the identical assimilation setup with a different land surface model.

Mean increments showed distinctive patterns with slight positive biases up to 1 % soil moisture in areas covered by denser vegetation and neutral to slightly negative impact for areas mostly covered by sparse vegetation. A possible cause could be the use of climatological LAI data, which is common practice within current land data assimilation systems. Due to the abundance of operational available vegetation data we would like to encourage future studies to look into possible improvements by using

non-climatological LAI, where cloud cover permits. Climatological LAI might especially pose a problem for the monitoring of extreme events, such as droughts, since these tend to result in lower LAI values again influencing the forward simulations. Further, it might also be useful to add perturbations to the LAI data in order to better simulate the uncertainties of the forward simulations. It is known that remotely sensed LAI data is erroneous. McColl et al. (2011) for example show for the Murrumbidgee basin that MODIS MOD15A2 estimates are too high for lightly vegetated areas and too low for densely vegetated

areas. This coincides well with the contradicting patterns of this study with the the increments showing a slight positive bias for densely vegetated areas and vice versa. McColl et al. (2011) further describe the quasi gaussian distribution of the LAI uncertainties with a bias of -0.82 and a standard deviation of 0.82.

The dependence of SMOS soil moisture retrievals on landscape features, such as vegetation, within the Murrumbidgee basin was also shown by Su et al. (2013). The ability of explicitly accounting for these effects within the forward simulations and

to avoid cross-correlations with ancillary data used within the soil moisture retrieval is one of the advantages of the brightness temperature assimilation when compared to the assimilation of retrievals. Draper et al. (2009b) have evaluated AMSR-E soil moisture over Australia which correlates well with in-situ measurements. The seasonal soil moisture patterns also well reflect the ones observed within this study, most notably in the northern tropical regions as well as in the east. Draper et al. (2009b) argue that although vegetated areas mostly correspond to higher soil moisture, the retrievals might also be contaminated by the

vegetation signal.

Areas of denser vegetation are also mostly linked to higher precipitation. Seasonal variations for the increments are clearly linked to the seasonal precipitation patterns, and thus to the vegetation growing season. The multiplicative precipitation perturbations applied here have a large impact on the total simulated model uncertainty and the impact of the observations across

different geographic areas. Renzullo et al. (2014) applied an average multiplicative error of 60 % over Australia, closely matching the 50% applied in this study, for the BAWAP rain-gauge based precipitation data based on the analysis by Jones et al. (2009). Strong spatial variations exist for these precipitation estimates, largely influenced by the amount of locally available gauge stations. The ERA-Interim analysis data used within this study, produced by assimilating a multitude of both satellite as well as in-situ data, equally has errors linked to geographic areas. Estimates on these are however not provided with the

product. The updated ERA-5 reanalysis data, which will include information on ensemble mean and spread, could therefore be a significant step forward in characterising the background error induced by the meteorological forcings when using global, non locally optimised, data. López López et al. (2016) have assimilated AMSR-E soil moisture data across the Murrumbidgee basin using global precipitation data as well as locally optimised forcing data. Since the latter increases the open-loop accuracy, the positive assimilation effect here is actually reduced.

Within the broader context of Australian soil moisture analysis, a study comparing soil moisture output from several models as well as satellite products to in-situ data carried out by (Holgate et al., 2016) showed that SMOS retrievals are favourable across measurement sites, except for one with dense tropical vegetation. This is attributed to the likely advantage of their deeper penetration capability. The same study also shows that retrievals from ascending orbits perform best. These findings relate to our study, where soil moisture simulations were mostly improved across all measurement sites and the unscaled ascending brightness temperature acquisitions both showed a smaller bias towards the forward simulations and had a larger effect on the soil moisture analysis. However, Su et al. (2013) report that their comparison of SMOS retrievals to in-situ measurements within the Murrumbidgee basin showed that the descending orbits performed better. (Holgate et al., 2016) also show that similarities are largest within the group of the satellite retrievals and within the group of the different model outputs, with on average larger differences between models and retrievals. This further motivates to combine observations and model output in an optimal way through data assimilation specifically for Australia, as performed here. (Kumar et al., 2017) compare soil moisture from simple model output to complex land surface simulations, arguing that the latter perform better within Australia. The CLM land surface model is a fully physical based land surface scheme solving the energy and mass balance and provides all data required for the forward simulation of the brightness temperatures, allowing the full use of L-band brightness temperature observations.

This enables the correct temporal alignment between observations and forward simulations is especially important to achieve a high accuracy for the forward simulations. Here, we slightly simplified the forward simulations by merging all ascending or descending daily overpasses and computing the forward simulations at one common time. The temporal offset is thereby maximum 3 hours, which is within the temporal resolution of the forcing data. No artefacts were identified although we encourage a more precise temporal alignment within operational systems, as is mostly done. Soil moisture retrievals are only valid for one specific time instance and inter-daily variation of soil moisture can be considerable due to precipitation events. The assimilation of observations into a model thus is advantageous, since it allows for more correct daily estimates by averaging the model time steps, here 30 minutes.

CLM uses fixed soil layer-depths which is likely the most beneficial model structure for comparing the assimilation effects spatially, since the covariance between observations and state variable does not vary depending on spatially non-uniform layer depths, as is the case with some other land surface models. The validity of updating very deep layers with information derived from surface observations is however questionable. Therefore joint assimilation schemes also assimilating data from satellites such as GRACE are preferable, as was done by Tian et al. (2017).

Long term assimilation effects were analysed by estimating the cumulative distribution functions for each grid cell prior to and after assimilation. On average, lower quantiles are shifted towards wetter conditions and higher quantiles are slightly shifted to drier conditions, resulting in reduced analysis variability. Spatial patterns in the quantiles do however change significantly at different quantile levels. We have shown these exemplary for the 10 % quantile. Here, for the experiment using homogeneous soil texture perturbations across all layers, the root-zone soil moisture showed a strong reduction compared to the open-loop run at the 10 % quantile level. Patterns visible in the increment bias were strongly exaggerated, highlighting the

problem of too large updates within the root zone and the general sensitivity towards model perturbations.

The uncalibrated forward operator and the therefore necessary rescaling of the observations might be one possible cause for the reduced analysis variability as well as the spatial patterns. The observation rescaling is especially a challenge around the very low or very high values. The number of samples within these regions of the CDFs is small and the observations are contaminated with errors, which might not be zero on average. In this case the tails of the observation distribution will not represent the true maximum and minimum values. Furthermore, the standard parameterization of the forward operator is certainly not perfect. The spatio-temporal patterns linked to geographic areas or specific land cover classes, such as areas of higher vegetation, could likely be reduced by calibrating the forward observation model towards the observations at a grid cell level. This however comes with its own problems, for instance a possible reduction of the sensitivity towards soil moisture or several parameter sets achieving the same result (i.e. equifinality).

Hydrological monitoring systems, where it is important to identify the relative occurrence of certain soil moisture levels and to monitor patterns both over space and time, are more and more likely to incorporate the assimilation of brightness temperatures with sufficiently long data records becoming available. Draper and Reichle (2015) have shown that data assimilation is able to correct modelled soil moisture also at longer time intervals from sub-seasonal to seasonal scale and seasonal differences in the assimilation effect are reported across many studies, as also shown here. Existing hydrological monitoring systems, such as the US. drought monitor (Svoboda et al., 2002), the African Flood and Drought Monitor (Sheffield et al., 2014), the German Drought Monitor (Samaniego et al., 2013) or the Australian Water Resource Assessment (Van Dijk et al., 2011; Vaze et al., 2013) all use soil moisture quantiles at grid cell level to characterise different levels of severity and facilitate the comparison of soil moisture levels between grid cells. We have shown that the assimilation induced quantile changes will have an effect on the spatio-temporal classification of areas above or below a certain quantile level, although the characteristics of these changes will be highly dependent on the model and data assimilation system. Hopefully, assimilation will benefit the monitoring and analysis of future severe events, such as the Millennium drought in Australia (van Dijk et al., 2013). To model the complex feedback processes between soil moisture and vegetation is likely best performed using raw brightness temperatures and the therefore use of consistent data between the land surface model and the forward simulations.

In this paper, a relatively long time series of SMOS brightness temperatures has been assimilated into the Community Land Model across the Australian continent and soil moisture simulations are improved for the very largest part of in-situ measurements, both for top-layer and root-zone soil moisture. Finally, the Community Land Model is part of the Community Earth System Model and the here presented data assimilation system will in future also enable the analysis of the long-term impact of L-band brightness temperature assimilation within coupled land-atmosphere experiments.

*Author contributions.* DR carried out all presented work and wrote the draft version of the publication. XH gave continuous support concerning the data assimilation system and the many code changes required for this work. HL gave input on the technical design of the data

assimilation experiments as well as the evolving manuscript. CM and NV provided additional remarks and suggestions for the evolving manuscript.

*Acknowledgements.* The research presented in this paper is funded by BELSPO (Belgian Science Policy Office) in the frame of the
5   STEREO III programme – project HYDRAS+ (SR/00/302). The assimilation experiments were run on the JURECA supercomputer at Jülich Forschungszentrum. The open-source assimilation system DasPy is available at https://github.com/daspy/daspy and was adapted to meet the requirements of this study. Hans Lievens is a postdoctoral research fellow of the Research Foundation Flanders (FWO). The authors thank everyone involved in the OzNET and CosmOZ soil moisture networks as well as the maintainers of the International Soil Moisture Network.

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

**Table 1.** CLM soil layer depths and relative layer thickness in respect to sum of the two top layers. The relative thickness was used as a scaling factor for the soil perturbations, effectively decreasing ensemble spread and error covariance for lower levels.

| Layer Depth [m] | 0.018 | 0.045 | 0.09 | 0.17 | 0.290 | 0.493 | 0.829 | 1.383 | 2.296 | 3.802 |
|---|---|---|---|---|---|---|---|---|---|---|
| Perturb. scaling | 1 | 1 | 1 | 0.60 | 0.36 | 0.22 | 0.13 | 0.08 | 0.05 | 0.03 |

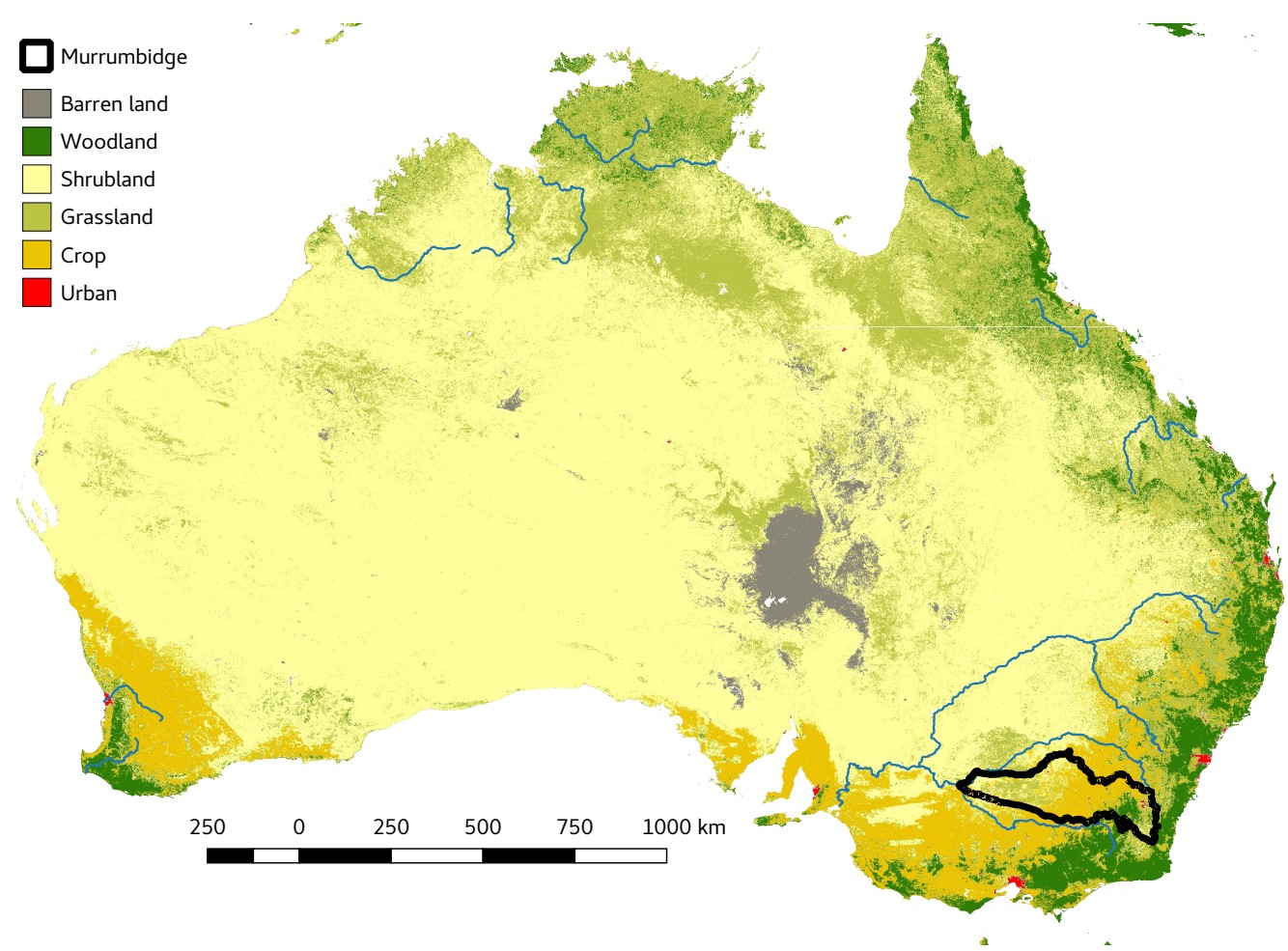

**Figure 1.** CLM plant functional types based on MODIS MCDQ12 land cover classification and ECOCLIMAP climate zones at 500 m resolution prior to the aggregation to 0.25 degrees. Some classes are here aggregated for visualisation purpose, e.g. evergreen temperate and evergreen tropical forests are both shown as Woodland. The boundary of the Murrumbidgee catchment, which is the site of the OzNet in-situ measurements, is shown.

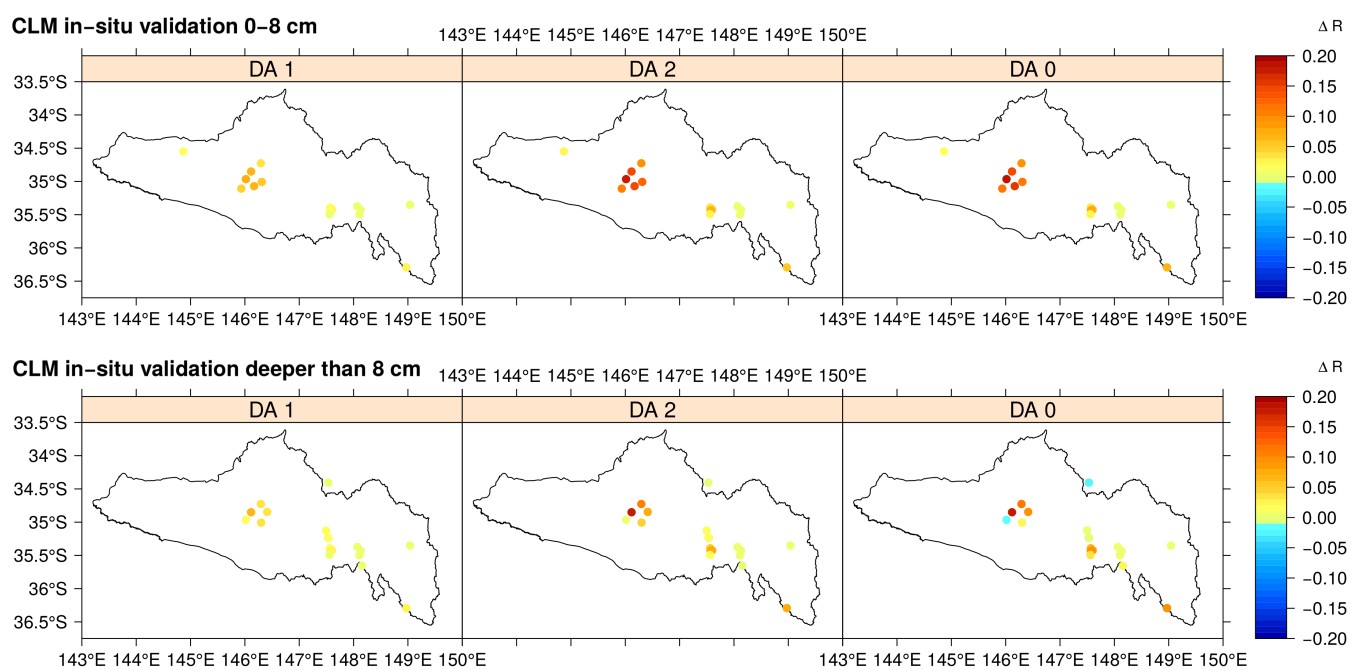

**Figure 2.** Change in correlation R for experiments DA 1, DA 2 and DA 0 both for top layer soil moisture (top panel) as well as the root-zone soil moisture (bottom panel) within the Murrumbidgee catchment. In the case of multiple measurements at the same location, the weighted average of the measured as well as modelled soil moisture was computed in accordance to the corresponding CLM layer thickness.

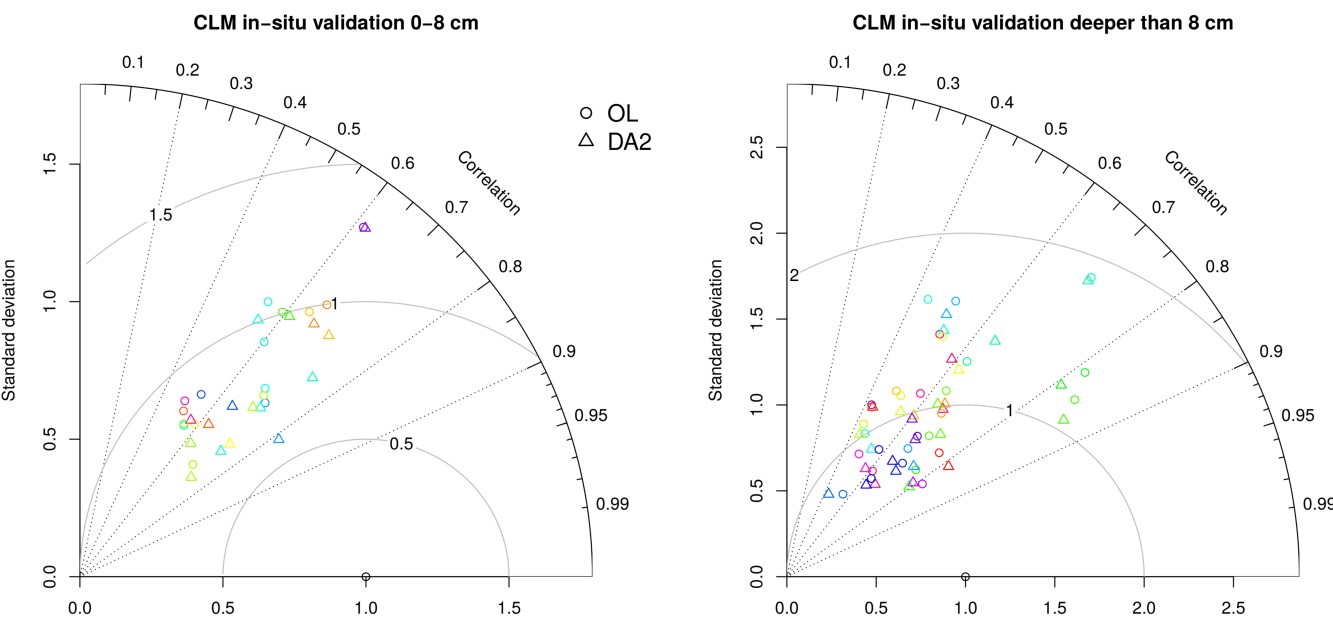

**Figure 3.** Taylor diagram showing assimilation impact on top layer soil moisture, defined as 8 cm soil depth, (left) and lower level soil moisture (right) in terms of correlation coefficient R, standard deviation and normalised RMSE for all in-situ measurement sites. Measurements at multiple depths are not aggregated.

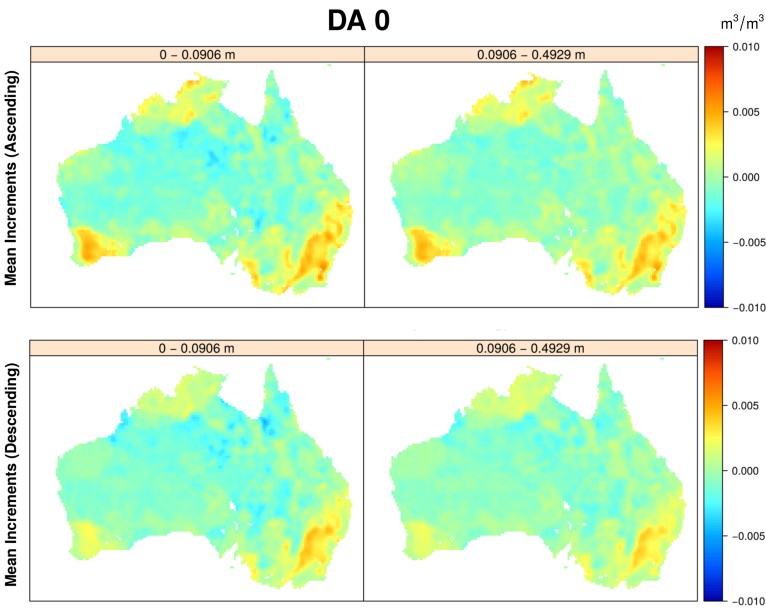

**Figure 4.** Mean of all increments for experiment DA 0 for top-layer soil moisture (left) and root zone soil moisture (right) for ascending (above) and descending orbits (below).

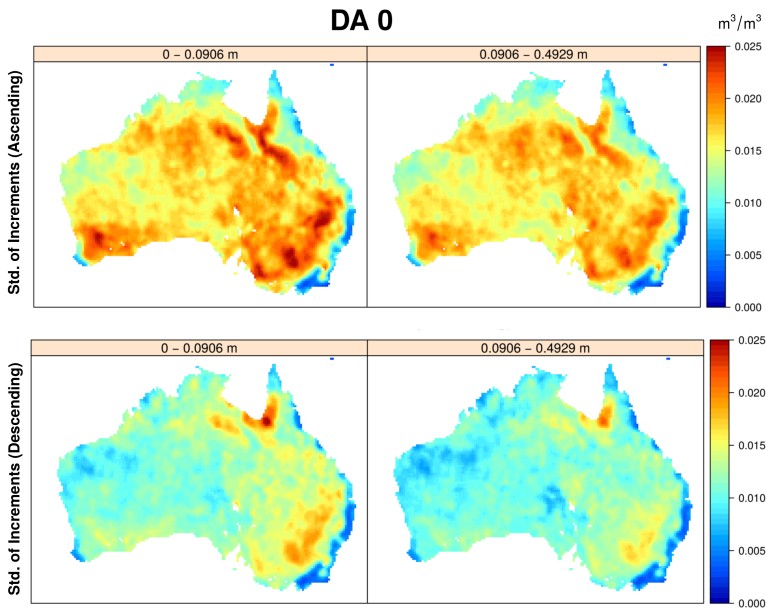

**Figure 5.** Increments standard deviation of all increments for experiment DA 0 for top-layer soil moisture (left) and root-zone soil moisture (right) for ascending (above) and descending orbits (below). Increments for the root-zone soil moisture are fairly similar to the top soil layers, due to the homogeneous texture perturbations applied across all layers.

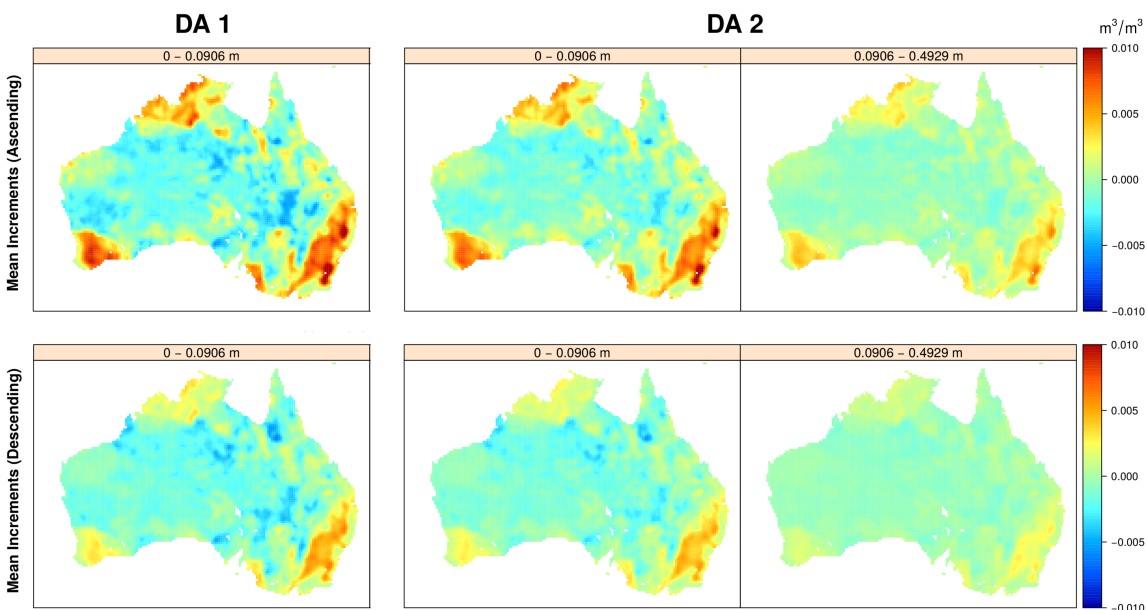

**Figure 6.** Mean of all increments for experiment DA 1 / DA 2 for top-layer soil moisture and root zone soil moisture for ascending (above) and descending orbits (below). Biases are strongest for the ascending orbit and distinctive spatial patterns are visible. Biases are strongly reduced both for deeper layers and descending orbits.

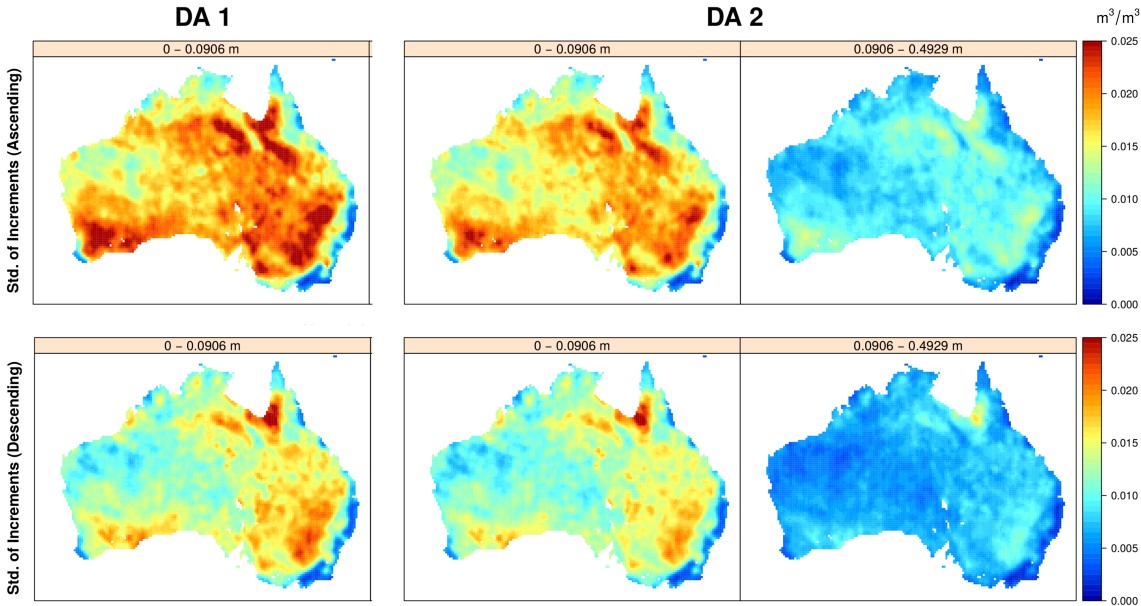

**Figure 7.** Standard deviation of increments for experiment DA 1 / DA 2 for top-layer soil moisture and root zone soil moisture for ascending (above) and descending orbits (below). Increments are strongest for the ascending orbit and for top-layer soil moisture and even stronger when restricting assimilation to these layers, as in DA 1. Increments are very low or zero for the forested areas along the coastline, either due to the absence of observations or the high LAI values masking any soil moisture signal within the forward operator.

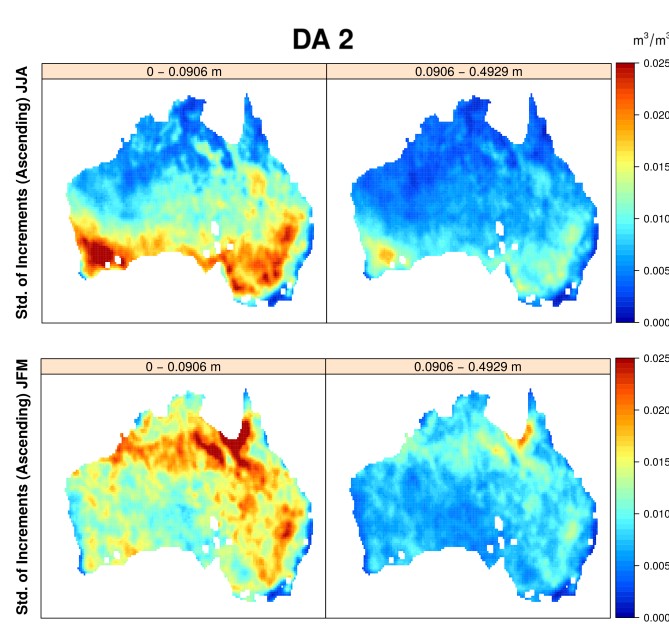

**Figure 8.** Increment standard deviation for ascending orbits for July - August (top) and January - March (bottom) for experiment DA 2. For the austral winter increments are strongest for the south of Australia, especially the agricultural areas. During austral summer the increments are strongest for the northern grasslands.

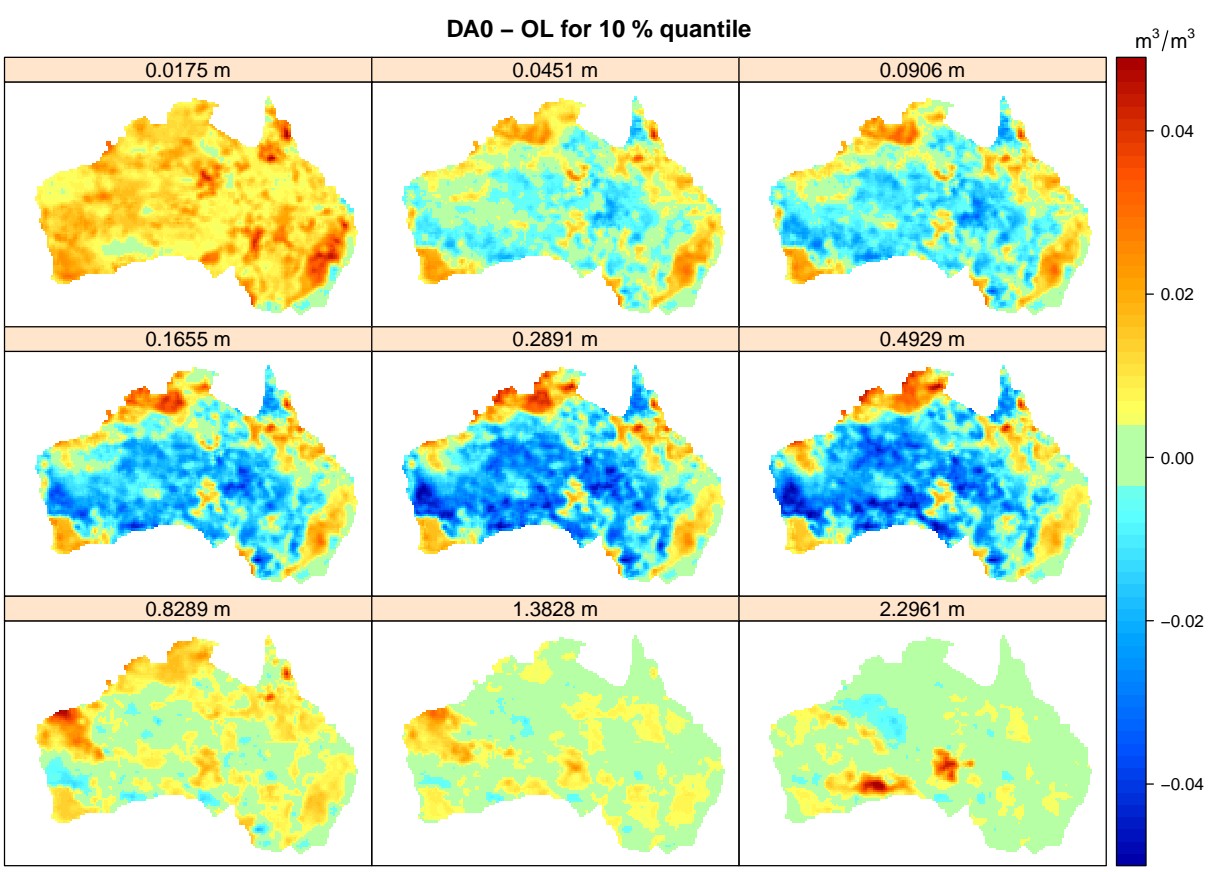

**Figure 9.** Differences in relative soil moisture between open-loop and DA 0 experiment (DA 0 - OL) for the 10 % quantile. The individual panels correspond to the top 9 CLM soil layers.

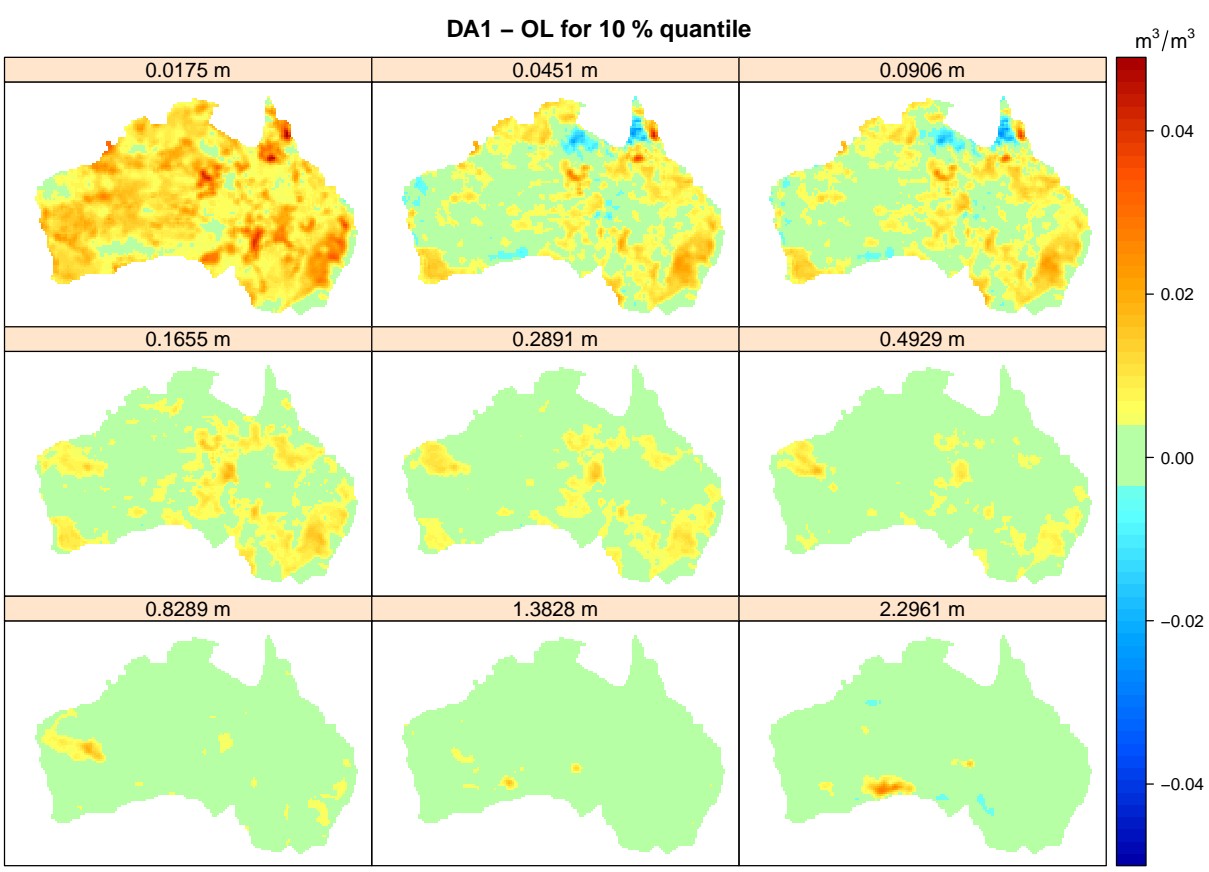

**Figure 10.** Differences in relative soil moisture between open-loop and DA 1 experiment (DA 1 - OL) for 10 % quantile. The individual panels correspond to the top 9 CLM soil layers titled with their depth.

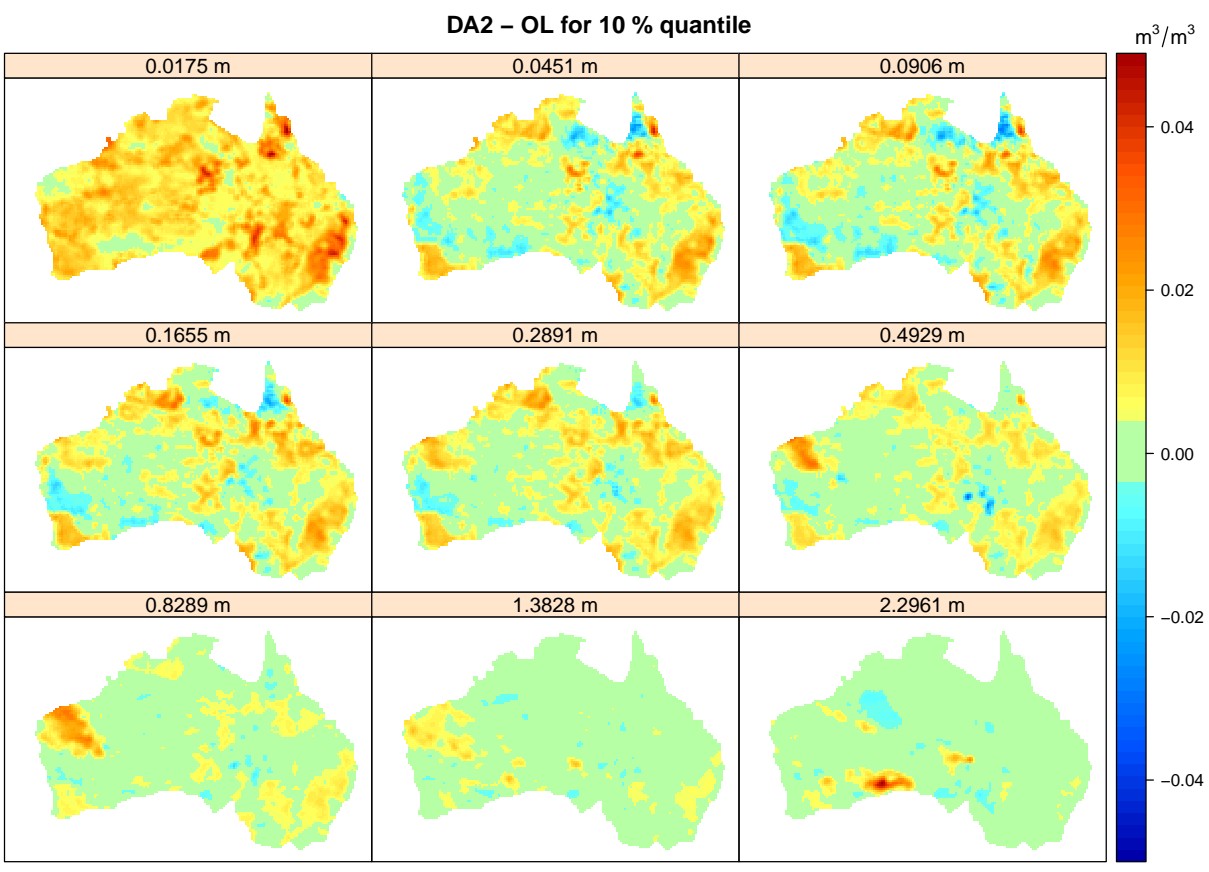

**Figure 11.** Differences in relative soil moisture between open-loop and DA 2 experiment (DA 2 - OL) for the 10 % quantile. The individual panels correspond to the top 9 CLM soil layers.

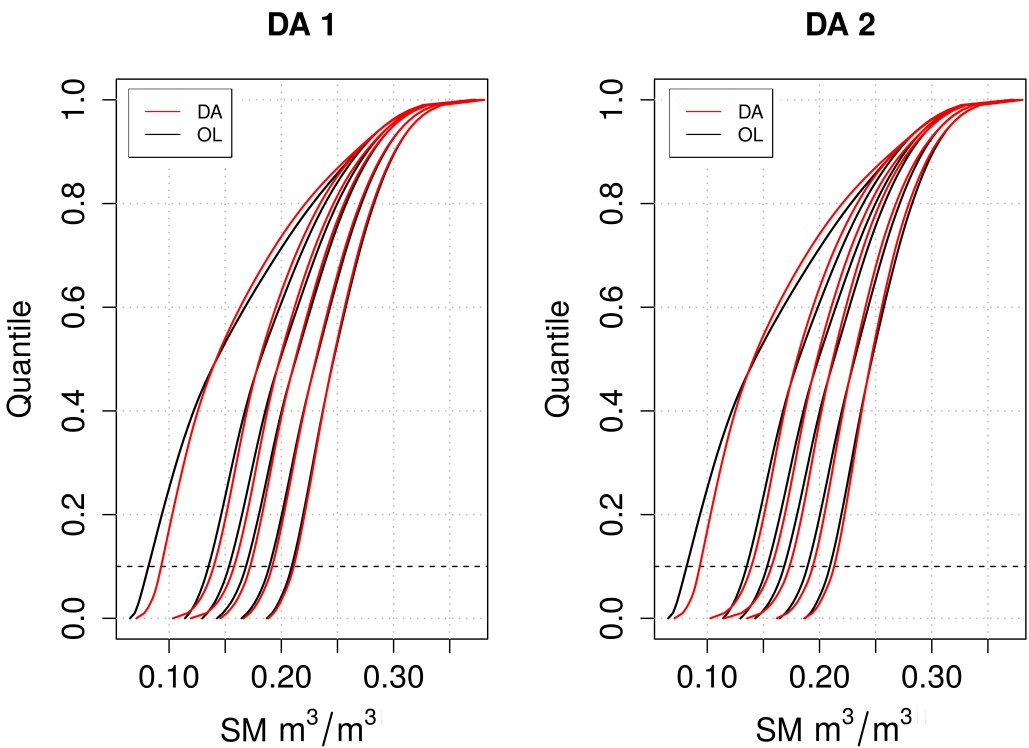

**Figure 12.** Cumulative distribution functions (CDFs) for the upper 6 CLM soil layers for experiments DA 1 and DA 2, based on quantiles computed for all data across the model domain. CDFs for open-loop simulations are shown in black and assimilation results in red. Both panels show changes in CDF behaviour for the layers being updated in the respective experiments, i.e. layers 1-3 for DA 1 and layers 1-6 for DA 2. Soil moisture increases systematically with soil depth allowing for the easy identification of the layers within the plot. The dashed vertical line marks the 10 % quantile, corresponding to figures 10 and 11.

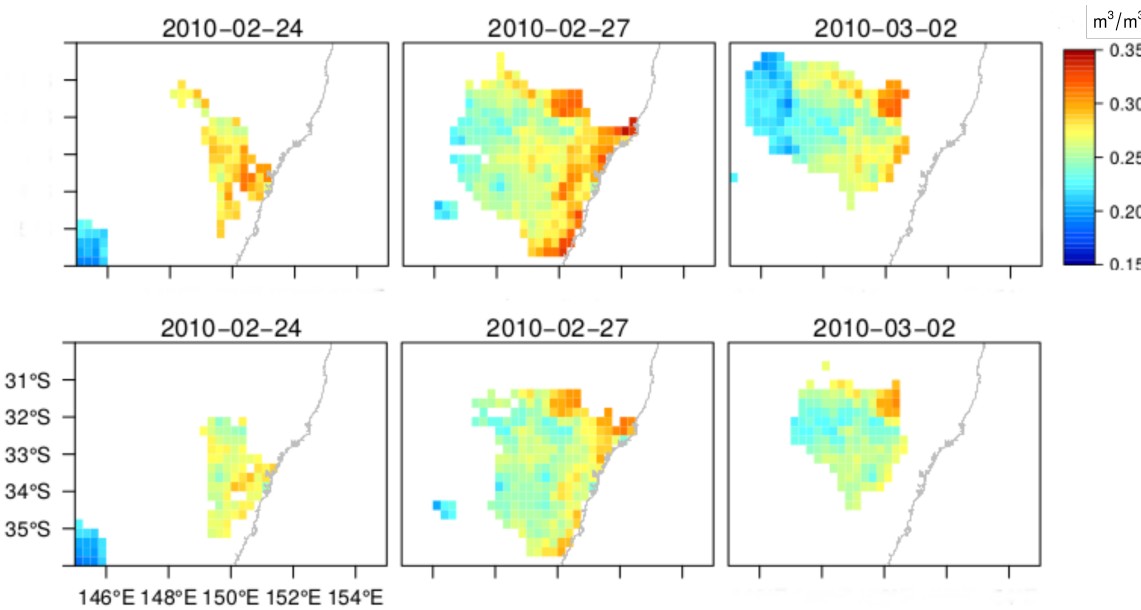

**Figure 13.** Sample drought event for February 2010, showing only the root zone soil moisture below the 10 % quantile level for the open-loop (above) and experiment DA 2 (below). The different spatial extent and differences in soil moisture itself, depending on the dataset used, at three different days are clearly visible. The figure is centred around the Central coast of New South Wales.