# Peer review of "SMOS brightness temperature assimilation into the Community Land Model"

_Hydrology and Earth System Sciences, 2017_

## Referee Comment (RC1) · Anonymous Referee #1 · 24 May 2017

Manuscript Number: hess-2017-188

Title: SMOS brightness temperature assimilation into the Community Land Model

Authors: Dominik Rains, Xujun Han, Hans Lievens, Carsten Montzka, and Niko E.C. Verhoest

**Major Comments**

This paper presents a data assimilation (DA) study where the SMOS brightness temperature is assimilated into the CLM model, forced with ERA-Interim surface meteorological fields, over the Australia area. The CMEM model is taken as the observation operator to simulate the $42.5^o$ incidence angle brightness temperature in H polarization and the LETKF algorithm from the DasPy package is used to perform the filter update.

[Figure]

The model ensemble is generated by perturbing both model parameters and forcing inputs. Three sets of DA experiments are carried out (DA1, DA2, DA0) with different numbers of soil layers included in the filter update and different ways to perturb the soil parameters. The filter updates are performed over brightness temperature anomalies (with seasonal cycle removed), which is different from most other studies. CDF matching is performed on the anomalies. Validations are carried out against ISMN in-situ observations. The results and analysis are focused on the soil moisture increments during the filter update and low soil moisture quantiles (10

This is a very carefully designed and carried out data assimilation study with its main novelty in assimilating brightness temperature anomalies. The investigation and results are significant and the quality of both the research and its presentation is very good – I see no major issues with the choices of the processing methods along the entire chain of DA procedures. The DA improvement, as measured by soil moisture skills (against ISMN), is reported as moderate, which is consistent with similar studies.

The discussions are relatively weak, especially on the effects of DA at different temporal scales. Draper and Reichle, 2015 decomposes the soil moisture time series into dynamics at different time scales (long-term, seasonal, and short-term) for the analysis. It is not exactly clear how (and why) the anomaly assimilation (which has the seasonal signals removed) changes the way the DA behaves at seasonal to longer time scales. Some time series plots and related analysis are needed to help on this. Also, the study area is very large and heterogeneous in terms of soil and vegetation – should there be any stratification on the analysis of the results, e.g., statistics over different types of soil/vegetation?

I think the paper can be published in HESS with minor revisions.

**Details**:

Page 9, line 6-7: the unites for observation errors are confusing – should they all be $K^2$ if they are all variances? Or they should all be in K if they are the standard deviation?

My guess is that they are all in K because $4^2 + 3^2 = 5^2$.

Figures 2, 4, 6, 7, 8: Maps here contain both negative and positive values and the sign of the data also matters. So it'll be much easier for the readers if a particular color (e.g. white) is used for the 0 values and two different sets of color shades (e.g. one set of warm shades and one set of cool shades) are used for positive and negative values.

Figures 6, 7, 8: What is [%/100]? Should it be just [%]? Change "0.1 quantile" to "10% quantile".

---

## Referee Comment (RC2) · L. J. Renzullo (Referee) · 30 May 2017

**GENERAL COMMENTS**

The paper explores a very interesting idea of assimilating brightness temperature observations, as opposed to derived soil moisture products, into a land surface model. However, no compelling argument is provide as to why this might be 'better' than assimilating the derived soil moisture product. Nor is there any real examination to the improvements, or otherwise, to the model performance. Most troubling however is the lack of evaluation against local information about the continental water balance to see if the patterns the authors have identified may be corroborated with either independent data or research. Why choose Australia as a case study but ignore the very many articles about data assimilation for water balance over the country? More detail critique is

provided in the following.

I recommend major revision and another review.

MAJOR ITEMS: * Most concerning is that there appears to be no interest in gaining new insight about Australian hydrology or indeed assessing the validity of the model estimates beyond soil moisture. No papers referencing Australian sources on the continent's hydrology or water cycle, so how do you know if the results are any good. There are clearly patterns in the results that may or may not be known to Australian research community. A simple first check is to see if the results accord with those from http://www.bom.gov.au/water/landscape/. The limited evaluation against in situ data in the Murrumbidgee catchment is weak, including no mention of the sites locations, the depth measured, nor what is measured (e.g. volumetric water content or wetness, neutron count etc).

* Should mention in the introduction that while L-band on SMOS may be the first 'dedicated' mission for soil moisture, there has been a long history of data assimilation development in C-band soil moisture retrievals (AMSR-E,-2 and ASCAT for example) and SMAP is yet another L-band mission that is providing global coverage. Moreover you would be wise to cite work from research who have performed assimilation with an Australian focus (you are not the first) and you cannot ignore the rich legacy of work conducted in understanding Australian hydrology.

* Simulations appear to be made for layers 0-9 cm, however the L-band sees emissions from at (at best) 0-5cm. Comment on this disparity and the impact, if any, on simulated brightness temperatures.

SPECIFIC ITEMS: P1,L17: Change to ' . . . sensitive to 1.4 GHz electromagnetic emissions, measures . . . multi angular top-of-atmosphere . . .'

P1,L18: Delete 'influenced by, among others, surface soil moisture.' Sentence is too long and 'among others' doesn't make sense. Among other what?
P1,L22: I suggest including the key reference by Kumar et al. 2009 in the list which describes the mechanisms how top layer soil moisture assimilation can improve root-zone estimates in LSM's. [Kumar, S. V., Reichle, R. H., Koster, R. D., Crow, W. T., & Peters-Lidard, C. D. (2009). Role of Subsurface Physics in the Assimilation of Surface Soil Moisture Observations. Journal of Hydrometeorology, 10(6), 1534–1547. https://doi.org/10.1175/2009JHM1134.1]

P2,L11: Suggest rewording the sentence to: ". . . retrievals represent the optimum fits between simulated brightness temperatures and the . . . "

P2,L19: Modify: "Sources of uncertainty include atmospheric forcing, . . ."

P2,L32-33: Not clear what is meant here. Elaborate on the link between brightness temperature and 'qualitative' models. Why would this even be a consideration?

P3,L2: "Within this" should be the start of a new paragraph.

P3,L2-35: Very lengthy introduction to what this paper is about. Strongly suggest restructuring to be more clear in the lead up to Section 2 about what the objectives of this paper are. The paragraph should start: "In this paper we . . " and itemise the objectives. This will help the reader link the findings with the objective of the work.

P3,L19: Why Australia? This needs to be clearly articulated.

P3,L26-28: If the results will be evaluated over Australia then you should cite "Smith et al. (2012)". Yes, these data are part of ISMN, but should cite the official source. [Smith, A. B., Walker, J. P., Western, A. W., Young, R. I., Ellett, K. M., Pipunic, R. C., . . . Richter, H. (2012). The Murrumbidgee soil moisture monitoring network data set. Water Resources Research, 48(7), 1–6. https://doi.org/10.1029/2012WR011976]

P3,L33: Change "avoiding to large" to "minimising the impact of potential large".

P4,L6: So did you use coupled or uncoupled mode? Why mention these if you're not going to specify here why.

P4,L9: What are these more recent higher resolution data sets? Explain that these will be described in Section 2.1 and 2.2. Why 0.25 degree? What is CLM normally run at? 0.25 degrees is quite coarse for continental studies, makes me think why not extend to the whole world. That way t=you can use the whole ISMN and not just the tiny little southeast corner of Australia?

P4,L13-32: How do you know if the derived surface information is accurate for Australia? What local information/expertise have you consulted? There is A LOT of research work (none of which are cite here) that shows these MODIS products are not representative of truth in Australia, (let alone the soils information). I would accept that a global study may use inferior information because it is the only data available with global coverage, but because this investigation focuses on Australia, it must be addressed! If accuracy is not an issue for this investigation (because assimilation compensates for the model deficiencies, including parameterisation) than you should state it explicitly here.

P5,L26: Change to '. . . allows the coupling of different . . .'

P6,L15: Is that "K ensembles" or "K ensemble members"? Clarify.

P6,L22: I recommend "mapped" instead of "propagated". Propagated is only relevant to mapping through time (or space).

P7,L29: The UTC to local time conversion may work for eastern Australia but not central or western Australia. How big an impact do you think a 2 hour error in timing will make on simulations?

P8,L16-17: Modelled brightness temperature can be extremely sensitive to choices in h, the roughness parameter. How have you dealt with this? Perhaps through the bias correction? Explain.

P8,L24: Is RFI an issue over Australia? If so, where will it be most likely. If not, then say so.

P10,L2-3: You need to be more specific. These are the OzNet network in the southeast of Australian in a catchment called the Murrumbidgee, I presume. If so, confirm and cite the relevant work (Smith et al, 2012). If not, then you need to explaining where the in situ data are located, how deep they measure, etc.

P10,L18-19: I would have thought the innovations would be close to zero on average (in fact that is one of the tests to see if your filter is operating optimally). Do you mean innovation or increment? Clarify. Also, what are the units on the increment? They appear to highlight dryland agricultural areas, e.g. western Australian wheat belt. Can you comment on the patterns and their connection to the surface parameterisation?

P11,L19: Please comment on the strong positive features in Fig. 6 in the 0.8 - 2.3 m layers. They are clearly linked to features in the landscape. What can you say about them?

P11,L34-P12,L1: A more relevant way to "place these findings in the context of" hydrological monitoring systems is to compare with actual modelling system output. A simple web search shows you can gain a lot of information about water balance in Australia from http://www.bom.gov.au/water/landscape/ I strongly urge you to consider locally relevant information to assess your results.

P12,L10-13: How do you know it was a drought event? What other independent corroborating evidence supports this?

P12,L23-24: You should mention the coupling to CMEM, as CLM does not estimate brightness temperatures.

P13,L5-7: Agree with revisiting the use of LAI climatology. Recommend further than you examine the usefulness in Australia.

Figure 1: Why cant the two panels be compared? They should be able to be compared. The point need to be identified, otherwise why have them a s separate shapes and colours?

Figure2 2-8,10: Why no label on the colour bar? Insert units. ('Unitless' is acceptable)

Figure 10: Where are we looking. Consider a location diagram/inset or mention: "central coast of New South Wales." for example.
* * *

---

## Referee Comment (RC3) · Anonymous Referee #3 · 1 Jun 2017

This study investigated the benefit of integrating SMOS brightness temperature and the Community Land Model over Australia. Three different scenarios were performed to update different layers of soil moisture by the LETKF method. The results were evaluated using ground soil moisture measurements. Personally, I think this paper was well written. The organization was reasonable and the experimental design was clear. However, there were still some major issues need to be addressed before it can be considered for publication. A more systematic literature review on remote sensing data-land surface model assimilation need to be conducted. There are two groups of remotely sensed soil moisture (or brightness temperature) assimilation studies, one for soil moisture estimation typically through land surface models, and the other for runoff and streamflow prediction normally through catchment hydrologic models. The current

introduction mixed these two together, with a lack of detailed review on remote sensing constrained land surface modelling. The contribution of this study should be better articulated based on the review of the current progress on this topic. The authors discussed extensively on bias issue in the Introduction and Results sections, which I agree is an important issue; however, I did not see what is new in this study in addressing this issue. The CDF matching is a traditional approach with the advantage of removing relative bias. However, the problem is that it does not estimate and disaggregate the relative bias into model one and observational one. I did not see how this study addressed this issue. The design of the different DA experiments were not well justified. Technically, there is no problem to update all soil moisture layers through cross covariance, which should maximize the benefit of assimilating remotely sensed surface soil moisture by addressing the gross error accumulated in the deep soil moisture. So what was the point of just update the first 9 cm? It may be argued that updating only surface soil moisture could test the ability of the CLM to update the deep soil moisture by the model dynamics itself; however, I do not think a Kalman filter is the best choice to answer this question. The error in deep soil moisture is an accumulation of the error from the surface soil moisture and a smoother to assimilate the RS data to update both current and past surface soil moisture will have a better capacity on testing the capability of the model to update deep soil moisture through model dynamics. Besides, more in-depth analysis and discussions need to be added. For instance, what is the implication of the results from this study on the issues such bias? Whether the results is reasonable (and being improved after data assimilation) for the whole Australia? Also, I would suggest the authors to be careful in using the words "assimilate" and "update". It should be very clear through the paper that RS surface soil moisture was "assimilated" while different layers in the model were "updated". P2L31: Based on the review above, I cannot get to the conclusion that TB assimilation is under researched compared with soil moisture retrieval assimilation. P7L10-15: Why 32 ensembles? Why no spatial correlation was considered while most of the errors are known to be spatially correlated? How these error parameters are estimated/determined? 50% of rainfall is a lot,

[Figure]

I reckon. P10L1-5: A bit of details on the soil moisture measurements quality control.

---

## Author Comment (AC3) · 26 Jul 2017

We thank the referee for the helpful comments. Due to the numerous changes in the manuscript, the new version is uploaded as a separate file.

Major Comments
This paper presents a data assimilation (DA) study where the SMOS brightness temperature is assimilated into the CLM model, forced with ERA-Interim surface meteorological fields, over the Australia area. The CMEM model is taken as the observation operator to simulate the 42.5 o incidence angle brightness temperature in H polarization and the LETKF algorithm from the DasPy package is used to perform the filter update.

The model ensemble is generated by perturbing both model parameters and forcing inputs. Three sets of DA experiments are carried out (DA1, DA2, DA0) with different numbers of soil layers included in the filter update and different ways to perturb the soil parameters. The filter updates are performed over brightness temperature anomalies (with seasonal cycle removed), which is different from most other studies. CDF matching is performed on the anomalies. Validations are carried out against ISMN in-situ observations. The results and analysis are focused on the soil moisture increments during the filter update and low soil moisture quantiles

This is a very carefully designed and carried out data assimilation study with its main novelty in assimilating brightness temperature anomalies. The investigation and results are significant and the quality of both the research and its presentation is very good – I see no major issues with the choices of the processing methods along the entire chain of DA procedures. The DA improvement, as measured by soil moisture skills (against ISMN), is reported as moderate, which is consistent with similar studies. The discussions are relatively weak, especially on the effects of DA at different temporal scales. Draper and Reichle, 2015 decomposes the soil moisture time series into dynamics at different time scales (long-term, seasonal, and short-term) for the analysis. It is not exactly clear how (and why) the anomaly assimilation (which has the seasonal signals removed) changes the way the DA behaves at seasonal to longer time scales. Some time series plots and related analysis are needed to help on this. Also, the study area is very large and heterogeneous in terms of soil and vegetation – should there be any stratification on the analysis of the results, e.g., statistics over different types of soil/vegetation? I think the paper can be published in HESS with minor revisions.

We have discussed this quite a bit and in the end did not include details on the temporal effects of the data assimilation. The paper is already quite extensive. However, we can gladly do this if the referee further suggest to do so. In that case we would suggest something on the increments, as shown in the two example figures below. We did not include any time series of soil moisture itself, since this would be for one specific location and thus not convey too much information. A land cover map has been added to the publication and patterns seen in the increments and quantiles are related to some features in Australia.

[Figure]

**Figure 1.** Standard deviation of increments (left) and increment bias (right).

Details: Page 9, line 6-7: the unites for observation errors are confusing – should they all be $K^2$ if they are all variances? Or they should all be in K if they are the standard deviation? My guess is that they are all in K because $4^2 + 3^2 = 5^2$.
Thank you for pointing this out. Yes, we have clarified this.

Figures 2, 4, 6, 7, 8: Maps here contain both negative and positive values and the sign of the data also matters. So it'll be much easier for the readers if a particular color (e.g. white) is used for the 0 values and two different sets of color shades (e.g. one set of warm shades and one set of cool shades) are used for positive and negative values. We have adjusted the colourbars and expanded the mid green as the neutral zone.

Figures 6, 7, 8: What is [%/100]? Should it be just [%]? Change "0.1 quantile" to "10 % quantile".
We have changed the figures accordingly.

---

## Author Comment (AC4) · 26 Jul 2017

We thank Luigi Renzullo for the very in-depth review, providing feedback both on major points as well as numerous details. For clarity our responses are added to his original comments and highlighted in colour.

The changes to the manuscript are attached in an extra file, since changes are pretty extensive.

GENERAL COMMENTS

The paper explores a very interesting idea of assimilating brightness temperature observations, as opposed to derived soil moisture products, into a land surface model. However, no compelling argument is provide as to why this might be 'better' than assimilating the derived soil moisture product. Nor is there any real examination to the improvements, or otherwise, to the model performance. Most troubling however is the lack of evaluation against local information about the continental water balance to see if the patterns the authors have identified may be corroborated with either independent data or research. Why choose Australia as a case study but ignore the very many articles about data assimilation for water balance over the country? More detail critique is provided in the following. I recommend major revision and another review.

We have reworked the relevant text passages within the introduction, highlighting the advantages of the assimilation of brightness temperature as opposed to soil moisture products. The CLM land surface model used in this study provides all necessary dynamic information required for the brightness temperature forward simulations, e.g. soil temperature, vegetation temperature etc. Furthermore, the static surface datasets within of the model are also used within the forward simulations. In contrast to this, soil moisture products are based on retrievals using output and surface datasets from other models, thus introducing inconsistencies. However, brightness temperature assimilation does have its own issues, e.g. shortcomings in the forward simulations and biases between simulated and observed brightness temperatures.

We have compared the modelled soil moisture to in-situ stations over Australia and evaluated the assimilation performance in terms of the correlation coefficient R and the Root Mean Square Error (RMSE), as done in other studies. The validation has been expanded, giving some more detail on the in-situ measurements and citing the relevant publications. We have included maps showing the Murrumbidgee basin where most of the in-situ measurements are located as well as the Australian land cover classification used by the model. Spatial patterns for the increments and also quantile evaluation are put into context of the Australian landscape.

We have added some lines in the introduction about why Australia was chosen as a study area and highlighted the main soil moisture assimilation studies that we have found.

MAJOR ITEMS: * Most concerning is that there appears to be no interest in gaining new insight about Australian hydrology or indeed assessing the validity of the model estimates beyond soil moisture. No papers referencing Australian sources on the continent's hydrology or water cycle, so how do you know if the results are any good. There are clearly patterns in the results that may or may not be known to Australian research community. A simple first check is to see if the results accord with those from http://www.bom.gov.au/water/landscape/. The limited evaluation against in situ data in the Murrumbidgee catchment is weak, including no mention of the sites locations, the depth measublue, nor what is measublue (e.g. volumetric water content or wetness, neutron count etc).

Please see above response. Further we would like to add that the study strongly focuses on soil moisture, and therefore the wider Australian water balance has not been discussed. We have compared the spatial patterns of the soil moisture simulations to the above link and now mention this. A quick check did reveal that for instance evaporation fluxes change by roughly 5 %,

* Should mention in the introduction that while L-band on SMOS may be the first 'ded- icated' mission for soil moisture, there has been a long history of data assimilation development in C-band soil moisture retrievals (AMSR-E,-2 and ASCAT for example) and SMAP is yet another L-band mission that is providing global coverage. Moreover you would be wise to cite work from research who have performed assimilation with an Australian focus (you are not the first) and you cannot ignore the

rich legacy of work conducted in understanding Australian hydrology.

We have included example studies on the use of ASCAT and AMSR-E in the introduction as well as studies focusing on Australia.

* Simulations appear to be made for layers 0-9 cm, however the L-band sees emissions from at (at best) 0-5cm. Comment on this disparity and the impact, if any, on simulated brightness temperatures.

For all experiments the forward simulations use model output from the 10 CLM layers. These reach far deeper than where L-band emissions mostly originate from, as stated up to roughly 5 cm. The forward operator accounts for this and the simulations therefore are also only sensitive to model output of 0-5cm. We have clarified this within the description of the experiments.

SPECIFIC ITEMS: P1,L17: Change to ' . . . sensitive to 1.4 GHz electromagnetic emissions, measures . . . multi angular top-of-atmosphere . . .'
We have changed this.

P1,L18: Delete 'influenced by, among others, surface soil moisture.' Sentence is too long and 'among others' doesn't make sense. Among other what?
We have shortened the sentence accordingly.

P1,L22: I suggest including the key reference by Kumar et al. 2009 in the list which describes the mechanisms how top layer soil moisture assimilation can improve root-zone estimates in LSM's. [Kumar, S. V., Reichle, R. H., Koster, R. D., Crow, W. T., & Peters-Lidard, C. D. (2009). Role of Subsurface Physics in the Assimilation of Surface Soil Moisture Observations. Journal of Hydrometeorology, 10(6), 1534–1547. https://doi.org/10.1175/2009JHM1134.1]
The citation has been included.

P2,L11: Suggest rewording the sentence to: ". . . retrievals represent the optimum fits between simulated brightness temperatures and the . . . "
The sentence has been changed.

P2,L19: Modify: "Sources of uncertainty include atmospheric forcing, . . ."
We have changed this.

P2,L32-33: Not clear what is meant here. Elaborate on the link between brightness temperature and 'qualitative' models. Why would this even be a consideration?
We have removed the term 'qualitative' as we agree it was not clear. We simply meant that brightness temperature assimilation should be tested with different land surface models

P3,L2: "Within this" should be the start of a new paragraph.
A new paragraph has been inserted.

P3,L2-35: Very lengthy introduction to what this paper is about. Strongly suggest restructuring to be more clear in the lead up to Section 2 about what the objectives of this paper are. The paragraph should start: "In this paper we . . " and itemise the objectives. This will help the reader link the findings with the objective of the work.
We have revised the introduction and moved some parts to the conclusion part of the paper (relevance of findings for drought monitoring systems, ellaborations on the use of CDFs and quantiles for extreme event characterisations in a shortened form).

P3,L19: Why Australia? This needs to be clearly articulated.
We have added some sentences on the motivation of choosing Australia as a study area.

P3,L26-28: If the results will be evaluated over Australia then you should cite "Smith et al. (2012)". Yes, these data are part of ISMN, but should cite the official source. [Smith, A. B., Walker, J. P., Western, A. W., Young, R. I., Ellett, K. M., Pipunic, R. C., ... Richter, H. (2012). The Murrumbidgee soil moisture monitoring network data set. Water Resources Research, 48(7), 1–6. https://doi.org/10.1029/2012WR011976]
The citation has been included.

P3,L33: Change "avoiding to large" to "minimising the impact of potential large".
The sentence has been altered.

P4,L6: So did you use coupled or uncoupled mode? Why mention these if you're not going to specify here why.
We do not mention the coupled mode anymore

P4,L9: What are these more recent higher resolution data sets? Explain that these will be described in Section 2.1 and 2.2. Why 0.25 degree? What is CLM normally run at? 0.25 degrees is quite coarse for continental studies, makes me think why not extend to the whole world. That way t=you can use the whole ISMN and not just the tiny little southeast corner of Australia?
We have included the reference to the relevant section. The 0.25 degree resolution matches the SMOS observatiosn well, which is now stated in the text. CLM itself can be run at many resolutions, although usually coupled global simulations are quite a bit coarser than 0.25 degrees, e.g. 0.5 or 0.75 degrees. The motivation of using Australia has been included, see above.

P4,L13-32: How do you know if the derived surface information is accurate for Australia? What local information/expertise have you consulted? There is A LOT of research work (none of which are cite here) that shows these MODIS products are not representative of truth in Australia, (let alone the soils information). I would accept that a global study may use inferior information because it is the only data available with global coverage, but because this investigation focuses on Australia, it must be addressed! If accuracy is not an issue for this investigation (because assimilation compensates for the model deficiencies, including parameterisation) than you should state it explicitly here.
We agree we should have consulted more local expertise. However, despite focusing on Australia we did have future global applications in mind choosing the datasets. We have included a study on the validation of LAI within the Murrumbidgee area in the Assimilation and results section, linking visible patterns to possible LAI errors. Selecting MODIS data was also motivated by the fact that a clear rational exists in using these data as CLM Plant Functional Types and LAI values. The soil data used incorporates local information, albeit into a global product. Concerning the forcing datasets, we have consulted local expertise but did not find forcings at the requried spatial and temporal resolution.

P5,L26: Change to '. . . allows the coupling of different . . .'
We have changed this.

P6,L15: Is that "K ensembles" or "K ensemble members"? Clarify.
Changed to "K ensemble members."

P6,L22: I recommend "mapped" instead of "propagated". Propagated is only relevant to mapping through time (or space).
We have changed this.

P7,L29: The UTC to local time conversion may work for eastern Australia but not central or western Australia. How big an impact do you think a 2 hour error in timing will make on simulations?
We have included that the assumed error is justifiable, since the 2 hour mismatch is smaller than the temporal resolution of the forcings. We do agree the approach is not optimal.

P8,L16-17: Modelled brightness temperature can be extremely sensitive to choices in h, the roughness parameter. How have you dealt with this? Perhaps through the bias correction? Explain.

We have included that the roughness parameter is important, but calibrating the forward simulations towards the observations might actually deteriorate the sensitivity towards soil moisture. We therefore keep the original parameters for good variability and remove the bias through CDF-matching.

P8,L24: Is RFI an issue over Australia? If so, where will it be most likely. If not, then say so.
Australia is largely unaffected by RFI, we have added this information

P10,L2-3: You need to be more specific. These are the OzNet network in the southeast of Australian in a catchment called the Murrumbidgee, I presume. If so, confirm and cite the relevant work (Smith et al, 2012). If not, then you need to explaining where the in situ data are located, how deep they measure, etc.
We have added the reference and also added a brief description of the sites location and depths. The OzNet sites location is now shown in a map.

P10,L18-19: I would have thought the innovations would be close to zero on average (in fact that is one of the tests to see if your filter is operating optimally). Do you mean innovation or increment? Clarify. Also, what are the units on the increment? They appear to highlight dryland agricultural areas, e.g. western Australian wheat belt. Can you comment on the patterns and their connection to the surface parameterisation?
We argue that for most parts the increments are close to zero but that deviations do exist. They are given in vol % soil moisture, which has been added to the graphics. Within the text we now refer to the possible error in the LAI values or other possible reasons, such as irrigation for limited areas.

P11,L19: Please comment on the strong positive features in Fig. 6 in the 0.8 - 2.3 m layers. They are clearly linked to features in the landscape. What can you say about them?
We have linked the patterns to Lake Eyre and the Nullarbor plain, they are the result of strong increments accumulating in the lower layers. The Nullarbor plain for instance is very dry, and adding water in the deep layers with low temporal variability will lead to strong quantile changes.

P11,L34-P12,L1: A more relevant way to "place these findings in the context of" hydrological monitoring systems is to compare with actual modelling system output. A simple web search shows you can gain a lot of information about water balance in Australia from http://www.bom.gov.au/water/landscape/ I strongly urge you to consider locally relevant information to assess your results.
We have included the site, stating that CLM output was compared to the AWRA-L simulations to check for consistency.

P12,L10-13: How do you know it was a drought event? What other independent corroborating evidence supports this?
We now refer to the event as being relatively dry, it is to highlight the influence of quantile changes without making any quantitative evaluations on real droughts. This could be interesting for future research.

P12,L23-24: You should mention the coupling to CMEM, as CLM does not estimate brightness temperatures.
We now mention CMEM.

P13,L5-7: Agree with revisiting the use of LAI climatology. Recommend further than you examine the usefulness in Australia.
Quite likely a next study, restricted to a more local area. We will consult local expertise upfront.

Figure 1: Why cant the two panels be compablue? They should be able to be compablue. The point need to be identified, otherwise why have them a s separate shapes and colours?

This has been corrected for.

Figure2 2-8,10: Why no label on the colour bar? Insert units. ('Unitless' is acceptable)
Done

Figure 10: Where are we looking. Consider a location diagram/inset or mention: "central coast of New South Wales." for example.
The proposed text has been added.

---

## Author Comment (AC2)

We thank the referee for his comments and hope to have answered them as best possible. Since the changes to the manuscript are extensive, the new version is uploaded as a separate file.

This study investigated the benefit of integrating SMOS brightness temperature and the Community Land Model over Australia. Three different scenarios were performed to update different layers of soil moisture by the LETKF method. The results were evaluated using ground soil moisture measurements. Personally, I think this paper was well written. The organization was reasonable and the experimental design was clear. However, there were still some major issues need to be addressed before it can be considered for publication. A more systematic literature review on remote sensing data - land surface model assimilation need to be conducted. There are two groups of remotely sensed soil moisture (or brightness temperature) assimilation studies, one for soil moisture estimation typically through land surface models, and the other for runof and streamflow prediction normally through catchment hydrologic models. The current introduction mixed these two together, with a lack of detailed review on remote sensing constrained land surface modelling. The contribution of this study should be better articulated based on the review of the current progress on this topic. The authors discussed extensively on bias issue in the Introduction and Results sections, which I agree is an important issue; however, I did not see what is new in this study in addressing this issue. The CDF matching is a traditional approach with the advantage of removing relative bias. However, the problem is that it does not estimate and disaggregate the relative bias into model one and observational one. I did not see how this study addressed this issue. The design of the different DA experiments were not well justified. Technically, there is no problem to update all soil moisture layers through cross covariance, which should maximize the benefit of assimilating remotely sensed surface soil moisture by addressing the gross error accumulated in the deep soil moisture. So what was the point of just update the first 9 cm? It may be argued that updating only surface soil moisture could test the ability of the CLM to update the deep soil moisture by the model dynamics itself; however, I do not think a Kalman filter is the best choice to answer this question. The error in deep soil moisture is an accumulation of the error from the surface soil moisture and a smoother to assimilate the RS data to update both current and past surface soil moisture will have a better capacity on testing the capability of the model to update deep soil moisture through model dynamics. Besides, more in-depth analysis and discussions need to be added. For instance, what is the implication of the results from this study on the issues such bias? Whether the results is reasonable (and being improved after data assimilation) for the whole Australia? Also, I would suggest the authors to be careful in using the words "assimilate" and "update". It should be very clear through the paper that RS surface soil moisture was "assimilated" while different layers in the model were "updated". P2L31: Based on the review above, I cannot get to the conclusion that TB assimilation is under researched compared with soil moisture retrieval assimilation. P7L10-15: Why 32 ensembles? Why no spatial correlation was considered while most of the errors are known to be spatially correlated? How these error parameters are estimated/determined? 50% of rainfall is a lot, I reckon. P10L1-5: A bit of details on the soil moisture measurements quality control.

Literature review: We have added some assimilation case studies, some specifically focusing on the study area. Also, while referencing these we now mention the model that has been used. We agree that catchment models vs land surface models can be quite different but don't see a big problem in referencing these together, since the assimilation steps are usually quite similar.

CDF-matching: We agree that the introduction was too long (as also pointed out by referee 2). We have therefore shortened and streamlined it towards the objectives of the study. Quite some detail on observation rescaling has been removed, or when appropriate moved to other parts, since it might have caused the false impression of the issue being resolved within this study or that this study heavily focuses on it.

Experiment design: We hope to have clarified the design of the experiments within the introduction as well as the assimilation and results section. Concerning the udating into different layers, we argue that errors in the upper layers are actually best fed into the deeper layers through model physics. The experiment DA 2 however shows, that directly updating the root-zone leads to further improvements. Due to the high temporal variability of upper soil layers we believe the Kalman filter is the method of choice for close to surface soil moisture. We agree that for lower layers a smoother might be an interesting option, such as is used for the assimilation of GRACE data.

Analysis and Discussion: We have added more in-depth discussions on the in-situ validation, patterns in the increments as well as the quantile analysis. A land cover map as well as maps to show the in-situ validation within the Murrumbidgee catchment have been added. We have clarified that the consistent improvement of correlation with in-situ measurements makes us believe that the results are valid for all Australia, although the problem of sparse in-situ measurement sites remains.

Assimilate vs Update: We have substituted the wording "assimilate" with "updating" where appropriate.

We have changed the sentence stating that TB assimilation is under researched to that it is relatively new in practical terms.

Number of Ensembles: We have added that around 30 Ensembles is common for land data assimilation studies.

Spatial noise: The assimilation was performed in 1D, thus not requiring spatially correlated perturbations. This has been added in the text. The relevant references have been added from which the perturbation factors were taken, including the rainfall perturbations.

Quality control: We have removed the sentence, the quality control was carried out globally and actually no sites in Australia were affected.